# A connectomics-based taxonomy of mammals

Laura E Suarez[1,2]*, Yossi Yovel[3], Martijn P van den Heuvel[4], Olaf Sporns[5], Yaniv Assaf[3], Guillaume Lajoie[2], Bratislav Misic[1]*

[1]Montréal Neurological Institute, McGill University, Montreal, Canada; [2]Mila - Quebec Artificial Intelligence Institute, Montreal, Canada; [3]School of Neurobiology, Biochemistry and Biophysics, Tel Aviv University, Tel Aviv, Israel; [4]Center for Neurogenomics and Cognitive Research, Vrije Universiteit Amsterdam, Amsterdam, Netherlands; [5]Psychological and Brain Sciences, Indiana University, Bloomington, United States

**Abstract** Mammalian taxonomies are conventionally defined by morphological traits and genetics. How species differ in terms of neural circuits and whether inter-species differences in neural circuit organization conform to these taxonomies is unknown. The main obstacle to the comparison of neural architectures has been differences in network reconstruction techniques, yielding species-specific connectomes that are not directly comparable to one another. Here, we comprehensively chart connectome organization across the mammalian phylogenetic spectrum using a common reconstruction protocol. We analyse the mammalian MRI (MaMI) data set, a database that encompasses high-resolution ex vivo structural and diffusion MRI scans of 124 species across 12 taxonomic orders and 5 superorders, collected using a unified MRI protocol. We assess similarity between species connectomes using two methods: similarity of Laplacian eigenspectra and similarity of multiscale topological features. We find greater inter-species similarities among species within the same taxonomic order, suggesting that connectome organization reflects established taxonomic relationships defined by morphology and genetics. While all connectomes retain hallmark global features and relative proportions of connection classes, inter-species variation is driven by local regional connectivity profiles. By encoding connectomes into a common frame of reference, these findings establish a foundation for investigating how neural circuits change over phylogeny, forging a link from genes to circuits to behaviour.

**\*For correspondence:**
laura.suarez@mail.mcgill.ca (LES);
bratislav.misic@mcgill.ca (BM)

**Competing interest:** The authors declare that no competing interests exist.

## Editor's evaluation

This important article uses an impressively rich data set (obtained and curated by the authors) to compare the structural brain connectomes of many animals spanning six taxonomic orders. The approach is innovative and relies on graph theoretical measures to describe the connectivity, which means it can be done without the need to spatially/functionally match the brains. The authors find compelling evidence that there is more variability between than within order. They attribute this effect to changes in local connectivity features, whereas global patterns are preserved. The approach can potentially be a useful way to study phylogeny and brain evolution.

## Introduction

Anatomical projections between brain regions form a complex network of polyfunctional neural circuits (*Sporns et al., 2005*). Signalling on the brain's connectome is thought to support cognition and the emergence of adaptive behaviour. Advances in imaging technologies have made it increasingly

feasible to reconstruct the wiring diagram of biological neural networks. Thanks to extensive international data-sharing efforts, these detailed reconstructions of the nervous system's connection patterns have been made available in humans and in multiple model organisms (*van den Heuvel et al., 2016*), including invertebrate (*White et al., 1986*; *Chiang et al., 2011*; *Towlson et al., 2013*; *Worrell et al., 2017*), avian (*Shanahan et al., 2013*), rodent (*Oh et al., 2014*; *Bota et al., 2015*; *Rubinov et al., 2015*), feline (*Scannell et al., 1995*; de Reus and *de Reus and van den Heuvel, 2013*; *Beul et al., 2015*), and primate species (*Markov et al., 2012*; *Majka et al., 2016*; *Liu et al., 2020*).

The rising availability of connectomics data facilitates cross-species comparative studies that identify commonalities in brain network topology and universal principles of connectome evolution (*van den Heuvel et al., 2016*; *Barker, 2021*; *Barsotti et al., 2021*). A common thread throughout these studies is the existence of non-random topological attributes that theoretically enhance the capacity for information processing (*Sporns, 2013*). These include a highly clustered architecture with segregated modules that promote specialized information processing (*Watts and Strogatz, 1998*; *Hilgetag and Kaiser, 2004*), as well as a densely interconnected core of high-degree hubs that shortens communication pathways (*van den Heuvel et al., 2012*), promoting the integration of information from distributed specialized domains (*Zamora-López et al., 2010*; *Avena-Koenigsberger et al., 2017*). These universal organizational features suggest that connectome evolution has been shaped by two opposing and competitive pressures: maintaining efficient communication while minimizing neural resources used for connectivity (*Bullmore and Sporns, 2012*).

While comparative analysis can focus on commonalities among mammalian connectomes and identify universal wiring principles, it can also be used to systematically explore differences among connectomes that confer specific adaptive advantages. Indeed, despite commonalities, architectural variations are also observed even among closely related species (*Barker, 2021*). Factors such as the external environment, genetics. and distinct gene expression programs also account for diversity in neural connectivity patterns (*Martinez and Sprecher, 2020*). Subtle variations in connectome organization may potentially account for species-specific adaptations in behaviour and cognitive function.

But how does the connectome vary over phylogeny? Traditionally, mammalian taxonomies were built on morphological differences among species (*Darwin, 1959*). Besides physical commonalities, species within the same taxonomic group also tend to share similar behavioural repertoires (*York, 2018*; *Bendesky and Bargmann, 2011*; *Yokoyama et al., 2021*). Modern high-throughput whole-genome sequencing has further delineated phylogenetic links and relationships among mammalian species (*Murphy et al., 2021*; *Zoonomia Consortium, 2020*; *Álvarez-Carretero et al., 2021*; *Seehausen et al., 2014*). In addition to refining the overall classification of mammals, whole-genome comparative analyses have established the genetic basis of phenotypic variation across phylogeny (*Murphy et al., 2021*). Whether inter-species differences in the organization of connectome wiring conform to this taxonomy remains unknown. How do genes sculpt behaviour across evolution? Could speciation events in the genome leading to variations in connectome architecture be the missing link between genomics and behaviour? Rigorously addressing these questions is challenging due to the lack of methodological consistency in the acquisition and reconstruction of neural circuits, or the limited number of available species.

Here, we comprehensively chart connectome organization across the mammalian phylogenetic spectrum. We analyse the mammalian MRI (MaMI) data set, a comprehensive database that encompasses high-resolution ex vivo diffusion and structural (T1- and T2-weighted) MRI scans of 124 species (a total of 225 scans including replicas) (*Assaf et al., 2020*). All images were acquired using the same scanner and protocol. All connectomes were reconstructed using a uniform parcellation scheme consisting of 200 brain areas, including cortical and subcortical regions. Notably, the MaMI data set spans a wide range of categories across different taxa levels of morphological and phylogenetic mammalian taxonomies (*Assaf et al., 2020*). Specifically, it includes animal species across 5 different superorders (Afrotheria, Euarchontoglires, Laurasiatheria, Xenarthra, and Marsupialia) and 12 different orders (Cetartiodactyla, Carnivora, Chiroptera, Eulipotyphla, Hyracoidea, Lagomorpha, Marsupialia, Perissodactyla, Primates, Rodentia, Scandentia, and Xenarthra).

Taking advantage of the harmonized imaging and reconstruction protocols, we quantitatively assess the similarity of species' connectomes to construct data-driven phylogenetic relationships based on brain wiring. We compare these inter-species wiring similarities with conventional morphologically and genetically defined mammalian taxonomies. We determine the extent to which connectome topology

conforms to established taxonomic classes, and identify network features that are associated with speciation.

## Results

The MaMI data set consists of high-resolution ex vivo diffusion and structural (T1- and T2-weighted) MRI scans of 124 species. Since there is no species-specific template, all connectomes were reconstructed using a uniformly applied 200-node parcellation. Having equally sized networks facilitates graph comparison but also implies a lack of direct correspondence between nodes across species. However, because our focus is on the statistics of connectomes' topology, this does not impact our analyses. As the size of the network is kept constant across all species, voxel size is normalized to brain volume. *Figure 1* shows the distribution of connectomes across 10 mammalian orders (out of the 12 present in the data set). We focus on the 6 orders that contain 5 or more distinct species (within the Laurasiatheria and Euarchontoglires superorders); these include Chiroptera, Rodentia, Cetartiodactyla, Carnivora, Perissodactyla, and Primates, resulting in a total of 111 different animal species and 203 brain scans. A complete list of the animal species included in the data set is provided in *Figure 1—figure supplement 1*.

### Connectome-based inter-species distances

Similarity between species' network architectures is estimated using two network-based distance metrics: spectral distance, based on the eigenspectrum of the normalized Laplacian of the connectivity matrix (see *Figure 2—figure supplement 1*; *de Lange et al., 2014*), and topological distance, based on a combination of multiscale graph features of the binary and weighted connectivity matrices (*Figure 2—figure supplements 2 and 3* show the distribution of individual local and global graph features, respectively; *Rubinov and Sporns, 2010*). For completeness, *Figure 2—figure supplements 4 and 5* show the cumulative distribution of binary and weighted local features, respectively, for individual species. Both methods measure how similar the architectures of two connectomes are. To identify brain connectivity differences across species, we need to be able to analyse data in a shared frame of reference. The normalized Laplacian eigenspectrum and the graph features of the connectivity matrix allow us to translate connectomes into a common feature space in which they are comparable, despite the fact that they come from different species, and that the nodes do not correspond to one another (*Mars et al., 2021*). To account for the fact that some of the species have more than one scan, we randomly select one sample per species and estimate (spectral and topological) inter-species distances. We repeat this procedure 10,000 times and report the average across iterations.

*Figure 2a* shows the spectral distances between species' connectomes. In general, we observe smaller distances among members of the same order (outlined in yellow). *Figure 2b* confirms this intuition by showing that spectral distances within orders (i.e. values along the diagonal) tend to be smaller than distances between orders (i.e. values off the diagonal). *Figure 2c* shows the distributions of intra- and inter-order distances. The mean/median intra-order distance is significantly smaller than the mean/median inter-order distance (two-sample Welch's $t$-test: mean intra- and inter-order distances are 0.43 and 0.55, respectively, $p<10^{-4}$ two-tailed, and Cohen's $d$ effect size = 0.67; two-sample Mann–Whitney $U$-test: median intra- and inter-order distances are 0.44 and 0.55, respectively, $p<10^{-4}$ two-tailed, and common-language effect size = 68%; *Figure 2c*). We find comparable results when estimating species similarity using topological distance (two-sample Welch's $t$-test: mean intra- and inter-order distances are 0.41 and 0.53, respectively, $p<10^{-4}$ two-tailed, and Cohen's $d$ effect size = 0.59; two-sample Mann–Whitney $U$-test: median intra- and inter-order distances are 0.41 and 0.53, respectively, $p<10^{-4}$ two-tailed, and common-language effect size = 66%; *Figure 2d–f*). *Figure 2— figure supplement 6* shows the same results as in *Figure 2*, but using all samples including replicas (i.e. without random resampling). Altogether, results suggest that species with similar genetics, morphology, and behaviour tend to have similar connectome architecture. In other words, variations in connectome architecture reflect phylogeny.

### Architectural features differentiate species

Next we consider which network features contribute to the differentiation (*Figure 2—figure supplements 2 and 3* show the distributions of local and global graph features, respectively). To address

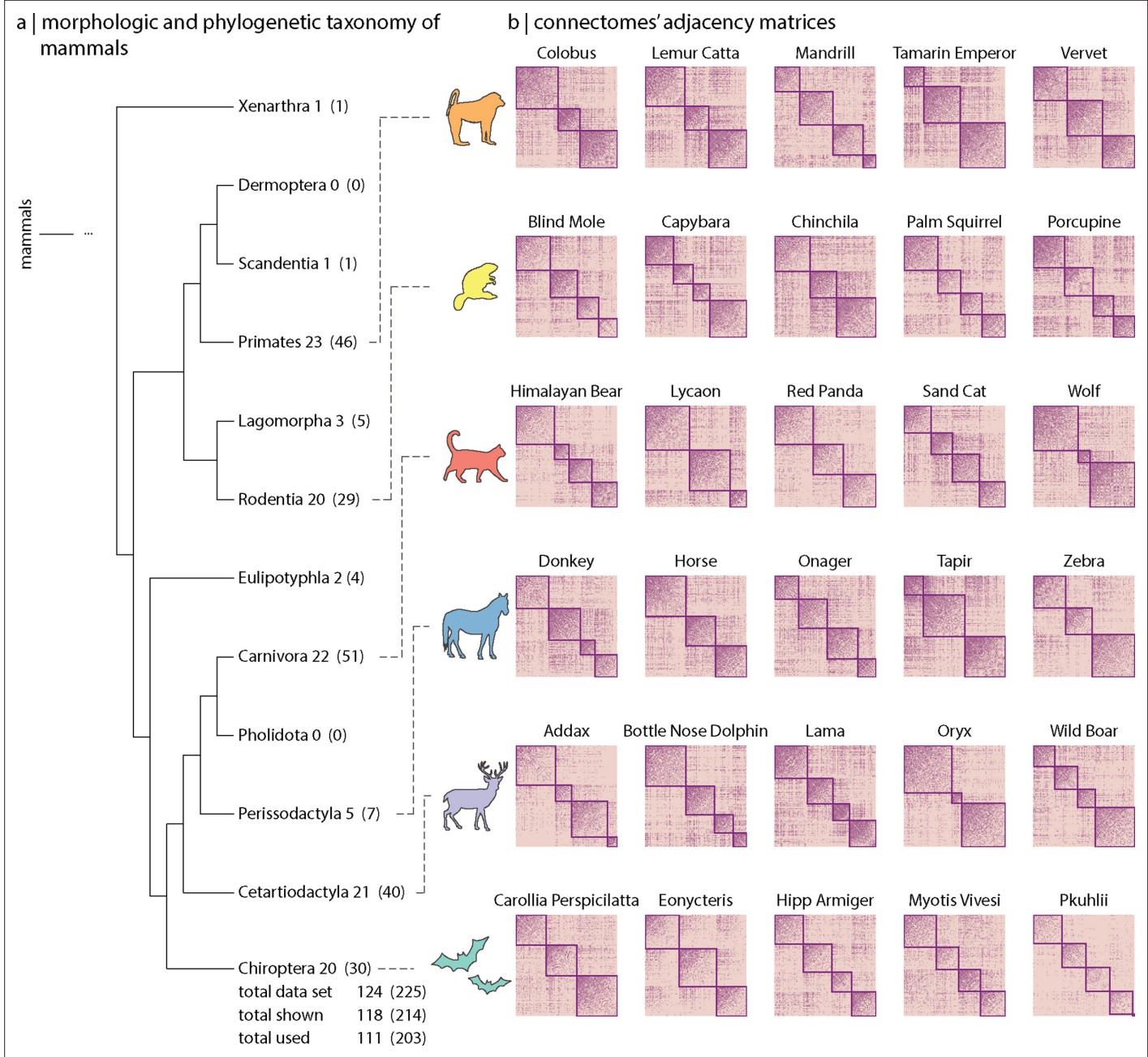

**Figure 1.** Mammalian MRI (MaMI) data set. The MaMI data set encompasses high-resolution ex vivo structural and diffusion MRI scans of 124 animal species spanning 12 morphologically and phylogenetically defined taxonomic orders: Cetartiodactyla, Carnivora, Chiroptera, Eulipotyphla, Hyracoidea, Lagomorpha, Marsupialia, Perissodactyla, Primates, Rodentia, Scandentia, and Xenarthra. (**a**) Hierarchical relationships across 10 (out of the 12 included in the data set) morphological and phylogenetic taxonomic orders. Numbers outside the parenthesis correspond to the number of unique species within each order, and numbers inside the parenthesis correspond to the number of samples (including replicas). (**b**) Connectivity matrices for five randomly chosen sample species within each of the six orders included in the analyses (i.e. Cetartiodactyla, Carnivora, Chiroptera, Perissodactyla, Primates, and Rodentia). Only orders with at least five different species were included for the analyses. Nodes are organized according to their community affiliation obtained from consensus clustering applied on the connectivity matrix (see 'Materials and methods'). Communities in (**b**) correspond to the partition for which the resolution parameter $\gamma = 1.0$ (*Figure 1—figure supplement 1*).

The online version of this article includes the following source data and figure supplement(s) for figure 1:

**Figure supplement 1.** Modularity.

**Figure supplement 1—source data 1.** List of animal species.

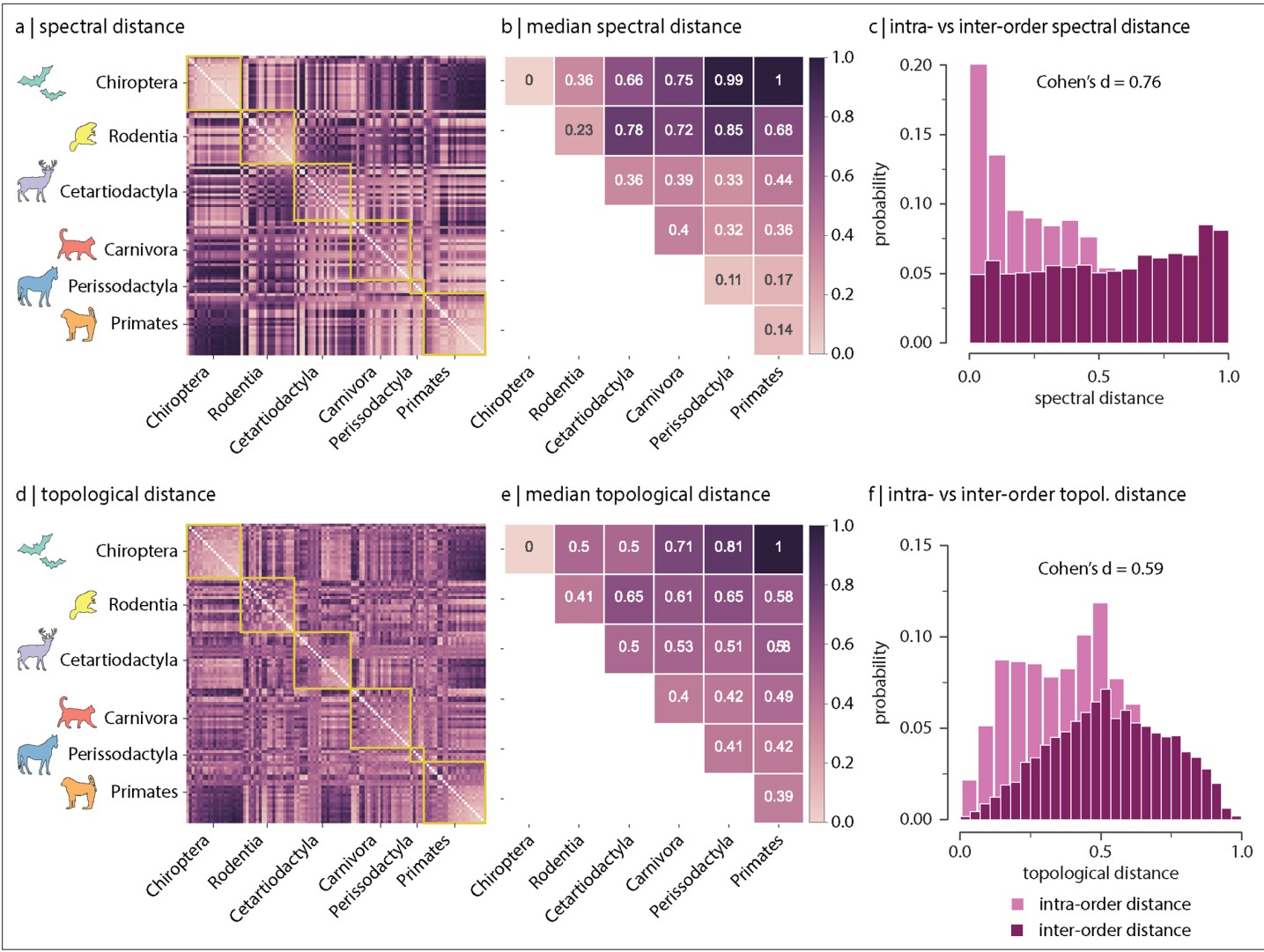

**Figure 2.** Spectral and topological distance between orders. (**a**) Spectral distance between species-specific connectomes. Lower distances indicate greater similarity. Yellow outlines indicate morphologically and genetically defined orders. (**b**) Median spectral distance within and between all constituent members of each order. (**c**) Distribution of intra- and inter-order spectral distances. (**d**) Topological distance between species-specific connectomes. Lower distances indicate greater similarity. Yellow outlines indicate morphologically and genetically defined orders. (**e**) Median topological distance within and between all constituent members of each order. (**f**) Distribution of intra- and inter-order topological distances. Effect sizes in (**c**) and (**f**) are Cohen's $d$ estimator corresponding to a two-sample Welch's $t$-test ($p < 10^{-4}$). Equivalent conclusions are drawn if common-language effect sizes from the two-sample Mann–Whitney $U$-test are used.

The online version of this article includes the following figure supplement(s) for figure 2:

**Figure supplement 1.** Laplacian eigenspectra.

**Figure supplement 2.** Distribution of local graph features across taxonomic orders.

**Figure supplement 3.** Distribution of global graph features across taxonomic orders.

**Figure supplement 4.** Cumulative distribution of binary local graph features across taxonomic orders.

**Figure supplement 5.** Cumulative distribution of weighted local graph features across taxonomic orders.

**Figure supplement 6.** Effect of using replicated samples on the topological and spectral distance between orders.

**Figure supplement 7.** Effect of (decreasing) parcellation resolution on the spectral and topological inter-species distance.

**Figure supplement 8.** Effect of (increasing) parcellation resolution on the spectral and topological inter-species distance.

**Figure supplement 9.** Effect of kernel density estimation (kde) on inter-species spectral distances.

this question, we recompute inter-species topological distances using different sets of graph features (*Figure 3*). We find that the difference between intra- and inter-order topological distances tends to be larger when only local (node-level) features are included in the estimation of the topological distance (i.e. degree, clustering coefficient, betweenness, and closeness; *Figure 3b and e*) compared to when only global features are considered (i.e. characteristic path length, transitivity, and assortativity; *Figure 3c and f*). This is the case for both the binary and weighted versions of these features (top and bottom rows in *Figure 3*, respectively). *Figure 3—figure supplement 1* shows the same results as in *Figure 3*, but using all samples including replicas (i.e. without random resampling). These results suggest that differentiation of orders is better explained by differences in local network topology; conversely, global network topology appears to be conserved across species. An illustration of this principle is depicted in *Figure 3—figure supplement 2* showing that the relative local connectivity of the anterior and the posterior ends of the cortex changes across taxonomic orders (*Barrett et al., 2020*; *Krubitzer and Kaas, 2005*; *Krubitzer and Kahn, 2003*).

A similar conclusion can be drawn when the eigenvalue distributions of the (normalized) Laplacian of the connectivity matrices are compared across species (*Figure 2—figure supplement 1*). In spectral graph theory, the presence of eigenvalues with high multiplicities or eigenvalues symmetric around $\lambda_i = 1$ provides information about the network's local organization that results from the recursive manipulation of connectivity motifs (*Banerjee and Jost, 2008*; *Banerjee and Jost, 2009*; *de Lange et al., 2014*). For instance, node duplication (i.e. the presence of nodes with the same connectivity profile) results in an increase of $\lambda_i = 1$. The duplication of edge motifs (i.e. the multiple presence of pairs of connected nodes with the same connectivity profile), on the other hand, produces eigenvalues at equal distances to $\lambda_i = 1$. Visually inspecting their Laplacian eigenspectra, one can notice that, across taxonomic orders, species tend to differ mostly around the interval $0.5 \leq \lambda_i \leq 1.5$, both in terms of the multiplicity of $\lambda_i = 1$, as well as in the width of the bell-shaped curve around $\lambda_i = 1$. While differences in the multiplicity of $\lambda_i = 1$ indicate differential amounts of duplicated node motifs present in the network, differences in the value and multiplicity of eigenvalues around $\lambda_i = 1$ indicate the presence of distinct edge motifs with disparate numbers of duplications in the network. Therefore, differences across taxonomic orders are most likely due to the presence of different local connectivity fingerprints in the connectivity matrix (*Figure 2—figure supplement 1*; *de Lange et al., 2014*; *Mars et al., 2018a*; *Mars et al., 2018b*). Determining which are specifically these node and edge motifs cannot be done by simply examining the Laplacian eigenspectra, and is out of the scope of this study. Additional evidence supporting the idea that spectral distance captures mostly differences in local network topology is the fact that the correlation between spectral and topological distance is maximum when only local features are included in the estimation of the topological distance (*Figure 3—figure supplement 3*).

We also observe that that the difference between intra- and inter-order topological distances is greater for binary than for weighted features (*Figure 3a–c and d–f*, respectively), independently of being local or global. This suggests that the strength of the connections is less important than the binary architecture of the connectivity matrix.

Some of the features used for the estimation of the topological distance depend on network density, which varies across taxonomic orders (*Figure 3—figure supplement 4*). To determine whether the observed differences between intra- and inter-order distances are above and beyond differences due to network density, we perform the same analysis shown in *Figure 3*, after controlling for density (*Figure 3—figure supplement 5*). Results, shown in *Figure 3—figure supplement 6*, suggest that differences between intra- and inter-order topological distances are not driven by differences in network density, but variations in wiring patterns, as captured by topological features, play a role in the observed phylogenetic variations in connectome organization.

Altogether, our results show that the subset of features that best differentiate species across taxonomic orders are the binary local topological features. We perform a set of complementary analyses to assess which subset of features produces the best partition of animal species relative to traditional taxonomies. To do so, we (1) project the data on a 2D plane using multidimensional scaling (*Figure 3—figure supplement 7*) and (2) apply hierarchical clustering to inter-species distance matrices (*Figure 3—figure supplement 8*). Visual inspection of these results suggests that, consistent with our previous results (*Figure 3*), local features compared to global features (ignoring panel a, centre vs. right column, respectively, in *Figure 3—figure supplements 7 and 8*), as well as binary

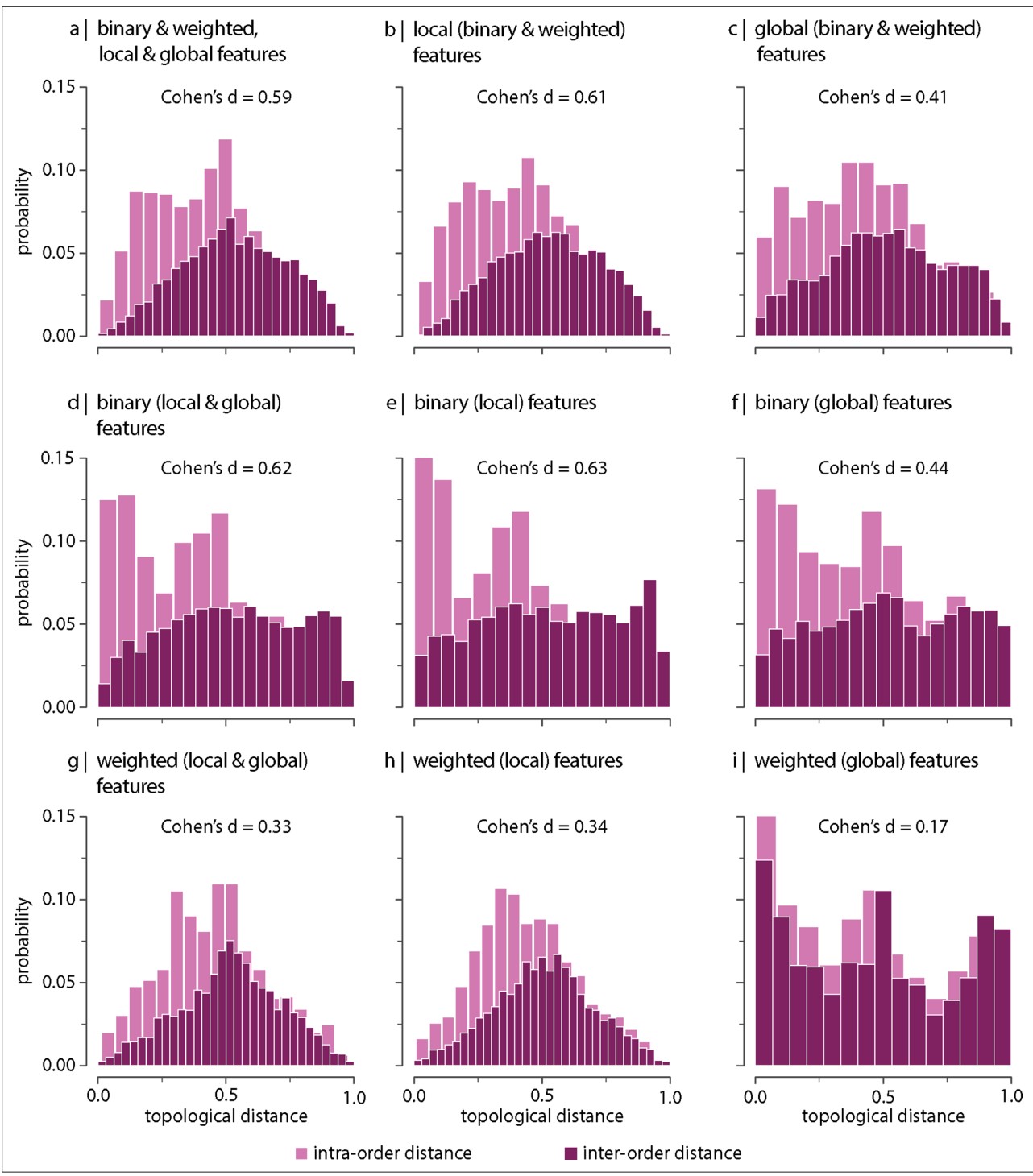

**Figure 3.** Contribution of network features. Topological distance can be computed using different combinations of local and global, binary and weighted connectome features. Histograms show intra- and inter-order distance distributions when using (**a**) all (binary, weighted, local, and global), (**b**) all local (binary and weighted), (**c**) all global (binary and weighted), (**d**) all binary (local and global), (**e**) only binary local, (**f**) only binary global, (**g**) all weighted (local and global), (**h**) only weighted local, and (**i**) only weighted global features. Local features include (the average and standard deviation of) degree, clustering, betweenness, and closeness. Global features include characteristic path length, transitivity, and assortativity. For definitions, please see 'Materials and methods.' Effect sizes correspond to Cohen's *d* estimator from a two-sample Welch's *t*-test. Equivalent conclusions are drawn if common-language effect sizes from a two-sample Mann–Whitney *U*-test are used. In all cases, the difference in the mean and median of intra- and inter-order distance distributions is statistically significant ($p<10^{-4}$). The same conclusions can be drawn after controlling for network density (*Figure 3— figure supplement 6*).

*Figure 3 continued on next page*

**Figure supplement 2.** Changes in local topology along the anterior–posterior axis.

**Figure supplement 3.** Relationship between spectral and topological distance.

**Figure supplement 4.** Network density.

**Figure supplement 5.** Controlling for network density.

**Figure supplement 6.** Contribution of network features after controlling for network density.

**Figure supplement 7.** Taxonomic order separation in low-dimensional space.

features compared to weighted features (ignoring panel a, centre vs. bottom row, respectively, in *Figure 3—figure supplements 7 and 8*), yield species partitions that more closely reflect established phylogenetic relationships, further supporting the idea that connectome organization recapitulates traditional taxonomic relationships that are based on morphology and genetics.

## Conservation of small-world architecture

Anatomical brain networks are thought to simultaneously reconcile the opposing demands of functional integration and segregation by combining the presence of functionally specialized clusters with short polysynaptic communication pathways (*Tononi et al., 1994*; *Sporns, 2013*; *Sporns et al., 2005*; *Bassett and Bullmore, 2006*). Such architecture is often referred to as small-world and is observed in a wide variety of naturally occurring and engineered networks (*Watts and Strogatz, 1998*). Here, we explore whether these principles of segregation and integration in global connectome organization are consistent across phylogeny. To do so, we estimate for each species the ratio of clustering coefficient to characteristic path length, normalized relative to a set of randomly rewired graphs that preserve the degree sequence of the nodes (*Humphries and Gurney, 2008*; *Maslov and Sneppen, 2002*; *Rubinov and Sporns, 2010*; *Figure 4*). Consistent with previous reports in individual species' connectomes (*Hilgetag and Kaiser, 2004*; *Sporns and Zwi, 2004*; *Bassett and Bullmore, 2006*), we find that all connectomes display high and diverse levels of small-worldness, suggesting that

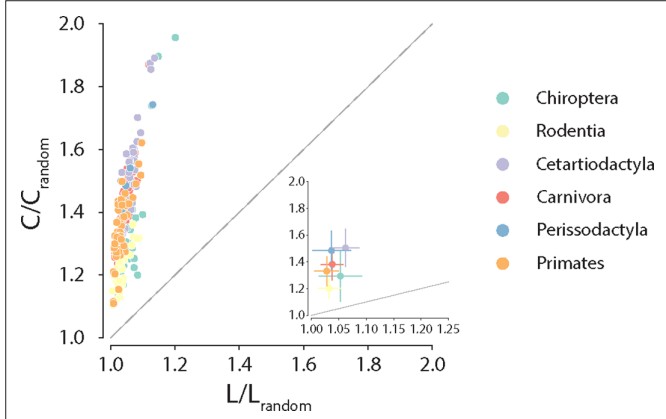

**Figure 4.** Conservation of small-world architecture. Clustering coefficient vs. characteristic path length normalized relative to a set of 1000 randomly rewired graphs that preserve the degree sequence of the nodes (*Maslov and Sneppen, 2002*). For definitions of each graph measure, see 'Materials and methods.' Each data point represents a different animal species. Data points above the identity line are said to have small-world architecture. The inset on the right bottom corner is a zoom on the abscissa; dots correspond to the median and error bars correspond to the standard deviation across species within the same taxonomic order.

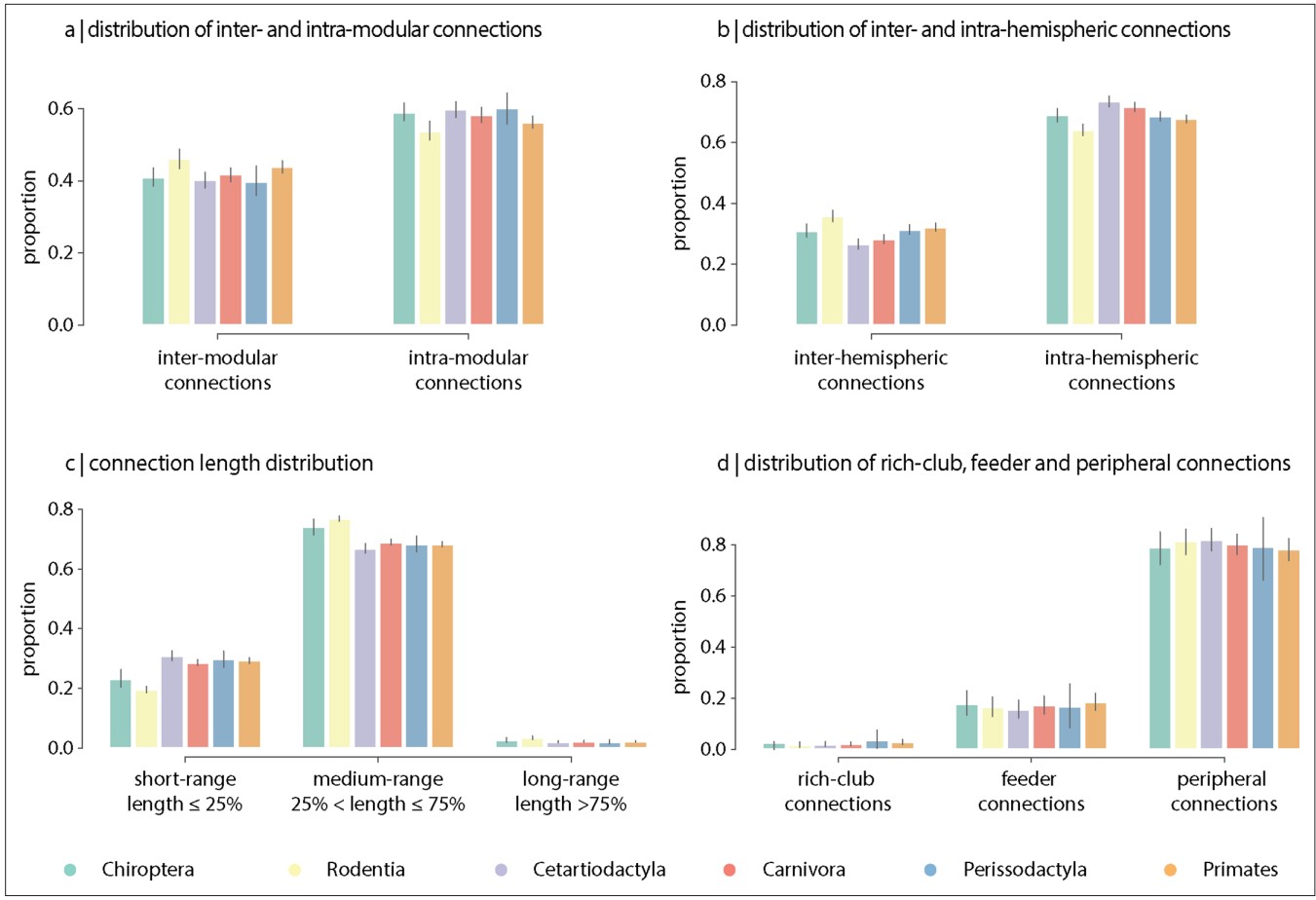

**Figure 5.** Contribution of edge types. Mean proportion of (**a**) inter- and intra-modular connections, (**b**) inter- and intra-hemispheric connections, (**c**) short- (length ≤ 25%), medium- (25% < length ≤ 75%) and long-range connections (length ≥ 75%), and (**d**) rich-club (connecting two rich-club nodes), feeder (connecting one rich-club and one non-rich-club node) and peripheral (connecting two non-rich-club nodes) connections. Error bars indicate 95% confidence intervals.

The online version of this article includes the following figure supplement(s) for figure 5:

**Figure supplement 1.** Connection length distribution.

simultaneously highly segregated and integrated networks is a global trait conserved across mammalian brains.

## Conservation of edge classes across species

The topological and spatial arrangement of connections in connectomes is thought to shape the segregation and integration of information and, ultimately, their computational capacity (*Faskowitz et al., 2021*). To investigate inter-species differences in the topological and spatial distribution of connections, we stratify edges into different classes in four commonly studied partitions. Partitions include inter- and intra-modular connections (*Figure 5a*), inter- and intra-hemispheric connections (*Figure 5b*), connection length distribution (short-, medium-, and long-range connections; *Figure 5c* and *Figure 5—figure supplement 1*), and rich-club (rich-club, feeder and peripheral connections; *Figure 5d*). Overall, we find that, along the four partitions, the relative proportions of each connection class are conserved across taxonomic orders, despite differences in connection density. Collectively, this is consistent with the results from the previous sections showing that global architectural features of connectomes are consistent across phylogeny.

## Discussion

In this study, we chart the organization of whole-brain neural circuits across 111 mammalian species and 5 superorders. We find that connectome organization recapitulates to a large extent traditional taxonomies. While all connectomes retain hallmark global features and relative proportions of edge classes, inter-species variation is driven by local regional connection profiles.

Conventional mammalian taxonomies are delineated based on the concept that a species is a group of organisms that can reproduce naturally with one another and create fertile offspring (*Mallet, 1995*). As a result, classical taxonomies based on animal morphology have largely been reconciled with emerging evidence from whole-genome sequencing *Baker and Bradley, 2006*; namely, organisms with similar genomes display similar physical characteristics and behaviour. Our work shows that inter-species similarity – as defined by morphology, behaviour, and genetics – is concomitant with the organization of neural circuits. Specifically, species that are part of the same taxonomic order tend to display similar connectome architecture, suggesting that brain network organization is under selection pressure (see also *Butler and King, 2004*; *Lande, 1976*; *Wright, 1931* for an alternative mechanism of trait evolution characterized by pure drift models based on Brownian motion), analogous to size, weight, or colour.

Which network features drive differences across taxonomic orders? Interestingly, all connectomes display consistent global hallmarks that were previously documented in tract-tracing studies, including high clustering and near-minimal path length characteristic of small-world organization, as well as segregated network communities and densely interconnected hub nodes (*van den Heuvel et al., 2016*). The conservation in global wiring and organizational principles is further supported by a reduced difference between intra- and inter-order topological distances estimated exclusively from global features compared to the case in which only local features are considered. Thus, relative differences between connectomes across taxonomic orders are mainly driven by local regional features. These results are in line with the idea that a brain region's functional fingerprint – the specific computation or function that it performs by virtue of its unique firing patterns and dynamics – is determined by its underlying cortico-cortical connectional fingerprint (*Mars et al., 2021*; *Mars et al., 2018b*; *Mars et al., 2016*; *Passingham et al., 2002*). Accordingly, inter-species differences in functional and behavioural repertoire are likely supported by changes in local connectivity patterns. Along the same lines, our results are also consistent with the notion that neural circuit evolution involves random local circuit modifications that may have provided species with behavioural adaptations, allowing them to face specific challenges (*Barker, 2021*), such as extreme environmental pressures *Park et al., 2008*; *Smith et al., 2011*; *Eigenbrod et al., 2019*, or to support specific behaviours, such as courtship (*O'Grady and DeSalle, 2018*; *Markow and O'Grady, 2005*; *Ding et al., 2019*; *Seeholzer et al., 2018*; *Khallaf et al., 2020*; *Barkan et al., 2018*; *Ding et al., 2016*; *York et al., 2019*), social bonding (*Insel and Shapiro, 1992*; *Winslow et al., 1993*; *Jaggard et al., 2020*; *Loomis et al., 2019*), or foraging (*Vanwalleghem et al., 2018*; *Pantoja et al., 2020*). How computations and cognitive functions emerge from these species-specific circuit modifications remains a key question in the field (*Buckner and Krienen, 2013*; *Suárez et al., 2021*).

These results highlight the importance of developing species-specific, anatomical-based parcellations, as well as new ways to align connectomes from different species. Understanding how homologous regions correspond to one another will allow further investigation of regional inter-species differences in connectome topology, which is a fundamental step for advancing comparative connectomics. A variety of emerging methods are contributing to further resolve correspondence between brain regions across species, facilitating fine-grained comparisons at the level of individual regions. These methods implement regional comparisons based on different data modalities including measures of cytoarchitecture (*J Garey, 1999*; *Bianchi et al., 2013*), receptor distribution (*Levant, 1998*), functional and structural connectivity fingerprints (*Passingham et al., 2002*; *Mars et al., 2016*), patterns of gene expression (*Warrington et al., 2022*; *Beauchamp et al., 2022*), and macroscale gradients of functional activation (*Buckner and Margulies, 2019*).

It is noteworthy that the relative proportion of edge classes (inter- vs. intra-modular, inter- vs. intra-hemispheric, short- vs. medium- vs. long-range and rich-club vs. feeder vs. peripheral) are preserved across species. This result is reminiscent of recent work on allometric scaling that investigates how white matter connectivity scales with brain size (*Bullmore and Sporns, 2012*). For example, species with fewer commissural inter-hemispheric connections exhibit lower hemispheric mean shortest path

(i.e. stronger intra-hemispheric connectivity), suggesting a similar connectivity conservation principle (*Assaf et al., 2020*). Likewise, using diffusion-weighted MRI data across 14 different primate species, another study reported negative allometric scaling of cortical surface area with white matter volume and corpus callosum cross-sectional area (*Ardesch et al., 2021*). This scaling results in less space for white matter connectivity with increasing brain size, translating into larger brains with a relatively higher proportion of short-range connections than long-range connections when compared with smaller brains (*Ardesch et al., 2021*). These results, however, do not contradict studies showing a positive allometric scaling between white matter and grey matter volume (*Zhang and Sejnowski, 2000*; *Theunissen, 1988*; *Schlenska, 1974*; *Frahm et al., 1982*); while the proportion of total volume devoted to cerebral white matter is higher in larger brains, it does not keep pace with the rapid cortical expansion that occurs with larger brain size. Collectively these studies highlight that the architecture of neural circuits and their physical embedding are intertwined, and the distribution of connections is such that it retains consistent global architectural features across phylogeny.

The present results contribute to the emerging field of comparative connectomics (*van den Heuvel et al., 2016*; *Mars et al., 2021*; *Barker, 2021*; *Tendler et al., 2021*). Adopting a harmonized imaging protocol in a large number of mammalian species facilitates a rigorous quantitative comparison of neural circuits. Central to this are network analytic methods that map connectomes to a common space and quantify similarities across local and global levels of organization (*de Lange et al., 2014*; *de Lange et al., 2016*; *Bassett and Sporns, 2017*; *Mars et al., 2018b*, *Mars et al., 2021*; *Warrington et al., 2022*). By comprehensively charting taxonomies of connectome architectures, we may uncover the principles that govern the wiring of neural circuits (*Avena-Koenigsberger et al., 2015*). In particular, quantitative analysis of connectome architecture across phylogeny may help to link genomics and behaviour (*Mišić and Sporns, 2016*). Traditionally, taxonomic groups were defined in terms of physical morphology and behavioural repertoire (*Burke, 1968*), but these are now understood to be driven by speciation events in the genome (*Murphy et al., 2021*; *Zoonomia Consortium, 2020*; *Álvarez-Carretero et al., 2021*). However, we do not yet understand how genes influence neural circuit architecture, which in turn shapes the behavioural repertoire of an organism. By understanding how neural circuits change over phylogeny, we can fill this gap and forge a link from genes to circuits to behaviour. Ultimately, the confluence of genomics, connectomics, and behaviour may help to triangulate towards a more well-rounded view of speciation (*Hernández-Hernández et al., 2021*), and their simultaneous investigation can further illuminate the link between structure and function in brain networks (*Bassett et al., 2010*; *Stiso and Bassett, 2018*; *Suárez et al., 2020*).

This work must be considered with respect to multiple limitations. First, uniformly parcellating brains of different size into the same number of nodes facilitates comparison of network architecture, but potentially obscures biologically important regional differences. Second, many species are represented by a single individual. Although we focus on orders rather than individual species and there is high within-species reliability (*Assaf et al., 2020*), the analyses do not capture individual variability within species. Third, all connectomes are reconstructed using diffusion-weighted imaging, which is subject to both systematic false positives and false negatives (*Maier-Hein et al., 2017*; *Schilling et al., 2019*). While a uniform, high-resolution ex vivo scanning protocol allows for systematic comparisons among species, and our results recapitulate findings from tract-tracing studies, future work in comparative connectomics will benefit from technological and analytical advances in neural circuit mapping. Fourth, evolutionary circuit modifications may not occur at the level of large-scale white matter, but at finer scales involving smaller nuclei or physiological events not accessible by diffusion imaging, such as up- or downregulation of neurotransmitter receptors (*Barker, 2021*). Nevertheless, the strikingly consistent taxonomic and phylogenetic relationships revealed by connectome analysis remain and suggest that macroscale connectivity, as measured by diffusion MRI, is informative of species similarities and differences across taxonomic orders.

By encoding connectomes into a common frame of reference, we quantitatively assess neural circuit architecture across the mammalian phylogeny. We find that connectome organization recapitulates previously established taxonomic relationships. Collectively, these findings set the stage for future mechanistic studies to trace the link between genes to neural circuits and ultimately to behaviour, and offer new opportunities to explore how changes in brain network structure across phylogeny translate into changes in function and behaviour.

## Materials and methods

### Brain samples

The MaMI database includes a total of 225 ex vivo diffusion and T2- and T1-weighted brain scans of 125 different animal species (*Figure 1—figure supplement 1*). No animals were deliberately euthanized for this study. All brains were collected based on incidental death of animals in zoos in Israel or natural death collected abroad, and with the permission of the national park authority (approval no. 2012/38645) or its equivalent in the relevant countries. All scans were performed on excised and fixated tissue. Animals' brains were extracted within 24 hr of death and placed in formaldehyde (10%) for a fixation period of a few days to a few weeks (depending on the brain size). Approximately 24 hr before the MRI scanning session, the brains were placed in phosphate-buffered saline for rehydration. Given the limited size of the bore, small brains were scanned using a 7-T 30/70 BioSpec Avance Bruker system, whereas larger brains were scanned using a 3-T Siemens Prisma system. To minimize image artefacts caused by magnet susceptibility effects, the brains were immersed in fluorinated oil (Flourinert, 3M) inside a plastic bag during the MRI scanning session.

### MRI acquisition

A unified MRI protocol was implemented for all species. The protocol included high-resolution anatomical scans (T2- or T1-weighted MRI), which were used as an anatomical reference, and diffusion MR scans. Diffusion MRI data were acquired using high angular resolution diffusion imaging (HARDI), which consists of a series of diffusion-weighted, spin-echo, echo-planar-imaging images covering the whole brain, scanned in either 60 (in the 7-T scanner) or 64 gradient directions (in the 3-T scanner) with an additional three non-diffusion-weighted images (B0). The b value was 1000 smm$^{-2}$ in all scans. In the 7-T scans, TR was longer than 12,000 ms (depending on the number of slices), TE was 20 ms, and Δ/δ = 10/4.5 ms. In the 3-T scans, TR was 3500 ms, with a TE of 47 ms and Δ/δ = 17/23 ms.

To linearly scale according to brain size the two-dimensional image pixel resolution (per slice), the size of the matrix remained constant across all species (128 × 96). Due to differences in brain shape, the number of slices varied between 46 and 68. Likewise, the number of scan repetitions and the acquisition time were different for each species, depending on brain size and desired signal-to-noise ratio (SNR) levels. To keep SNR levels above 20, an acquisition time of 48 hr was used for small brains (~0.15 ml) and 25 min for large brains (>1000 ml). SNR was defined as the ratio of mean signal strength to the standard deviation of the noise (an area in the non-brain part of the image). Full details are provided in *Assaf et al., 2020*.

### Connectome reconstruction

The ExploreDTI software was used for diffusion analysis and tractography (*Leemans et al., 2009*). The following steps were used to reconstruct fibre tracts:

1. To reduce noise and smooth the data, anisotropic smoothing with a 3-pixel Gaussian kernel was applied.
2. Motion, susceptibility, and eddy current distortions were corrected in the native space of the HARDI acquisition.
3. A spherical harmonic deconvolution approach was used to generate fibre-orientation density functions per pixel (*Tournier et al., 2004*), yielding multiple (n ≥ 1) fibre orientations per voxel. Spherical harmonics of up to fourth order were used.
4. Whole-brain tractography was performed using a constrained spherical deconvolution (CSD) seed point threshold similar for all samples (0.2) and a step length half the pixel size.

The end result of the tractography analysis is a list of streamlines starting and ending between pairs of voxels. Recent studies have shown that fibre tracking tends to present a bias where the vast majority of end points reside in the white matter (*Tournier et al., 2004*). To avoid this, the CSD tracking implemented here ensures that approximately 90% of the end points reside in the cortical and subcortical grey matter.

### Network generation and analysis

Before the reconstruction of the networks, certain fibre tracts were removed from the final list of tracts. These include external projection fibres that pass through the cerebral peduncle, as well as

cerebellar connections. Inner-hemispheric projections, such as the thalamic radiation, were included in the analysis. Brains were parcellated into 200 nodes using a *k*-means clustering algorithm. All the fibre end-point positions were used as input, and cluster assignment was done based on the similarity in connectivity profile between pairs of end points. Therefore, vertices with similar connectivity profile have a higher chance of grouping together. The clustering was performed twice, once for each hemisphere. Nodes were defined as the mass centre of the resulting 200 clusters. Connectivity matrices were generated by indexing the number of streamlines between any two nodes (*Assaf et al., 2020*). The resulting connectivity matrices are hence sparse and weighted adjacency matrices. For the analysis of the Laplacian eigenspectrum, connectivity matrices were binarized by setting connectivity values to 1 if the connection exists and 0 otherwise.

Even though the sizes of the regions differ across species, we opted for a uniform parcellation scheme (i.e. 200 nodes) for several reasons. First, to our knowledge, there is no MRI parcellation for the brains of the majority of the species studied here. Second, how brain regions correspond to one another across species (i.e. homologues) is still not completely understood for many regions and for many species. Third, comparing networks of different sizes introduces numerous analytical biases because most network measures trivially depend on size, making the comparison challenging. We therefore opted to implement a uniform parcellation scheme across species, allowing us to translate connectomes into a common reference feature space in which they can be compared (see 'Spectral distance' and 'Topological distance' sections). Note that this approach does not take into account species-specific regional delineations, nor does it capture homologies between nodes across species, which are still not completely understood. To ensure that the results are not idiosyncratic to the choice of parcellation and parcellation scale, we replicated all results using a lower resolution (100 node; *Figure 2—figure supplement 7* and *Figure 3—figure supplement 9*) and higher resolution (300 node; *Figure 2—figure supplement 8* and *Figure 3—figure supplement 10*) parcellation.

## Controlling for the scanning resolution and acquisition parameters

As the size of the matrix was kept constant across species (i.e. 200 nodes), voxel dimensions were linearly scaled with brain volume, thus resulting in different scanning resolutions across the samples. To verify that this was a reasonable assumption, several tests were performed: (1) the diffusion-based connectome of the mouse was previously compared against one derived from tract-tracing (see *Assaf et al., 2020* for details), obtaining a strong correlation between both networks. (2) Results on the connectivity conservation principle presented in *Assaf et al., 2020* were invariant to different scanning and parcellation parameters across nine different species. (3) The diffusion-weighted imaging method was able to reconstruct specific ground truth fibre systems across brains, and these fibre bundles scaled in size with brain volume.

## Spectral distance

To estimate similarities between species' connectome organization, we computed the Laplacian eigenspectrum of each graph. The Laplacian eigenspectrum acts like a spectroscopy of the graph and summarizes distinct aspects of the underlying topology (*Banerjee and Jost, 2009*; *Banerjee and Jost, 2008*; *Newman, 2001*; *Grone et al., 1990*; *Grone and Merris, 1994*; *Das, 2004*). We considered the normalized Laplacian matrix $L$ for undirected graphs with binary adjacency matrix $A$ defined as $L = I - D^{-1}A$, where $D$ is a diagonal matrix with $D(i,i) = \deg i$, and deg $i$ is the binary degree of vertex $i$.

$$
L(i,j) = \begin{cases} 1 & \text{if } i = j \\ -\frac{1}{\deg i} & \text{if } i \text{ and } j \text{ are connected} \\ 0 & \text{otherwise} \end{cases}
$$

with $i$ and $j$ representing two vertices of the graph. The Laplacian spectrum is then given by the set of all the eigenvalues of $L$. Importantly, the eigenspectrum of the normalized Laplacian has the advantage that all eigenvalues are in the range $[0, 2]$ (*Chung, 1996*), facilitating comparison across species. Furthermore, the normalized Laplacian is unitarily equivalent to the symmetric normalized Laplacian (*Chung, 1996*), that is, $L_{symm} = I - D^{-\frac{1}{2}}AD^{\frac{1}{2}}$, thus the eigenvalues of both Laplacians are real. The

spectral distance between every pair of animal species was then estimated as 1 minus the cosine similarity of their Laplacian eigenvalue distributions, where eigenvalue distributions were assumed to be vectors in a high-dimensional space.

To allow comparison of our results with previous reports (**de Lange et al., 2014**), in addition to comparing species using their connectome's Laplacian eigenspectra straightaway, we smoothed the eigenvalue distribution (i.e. $\lambda_1, \lambda_2, ..., \lambda_n$) by convolving eigenvalue frequencies with a Gaussian kernel. The new estimated density is given by

$$\Gamma(x) = \sum_{i=1}^{n} \frac{1}{\sqrt{2\pi\sigma^2}} \exp\left(-\frac{|x - \lambda_i|^2}{2\sigma^2}\right)$$

with $n$ being the number of eigenvalues in the approximated distribution, and $\sigma$ being a smoothing factor of 0.015. We used a step of 0.001, which resulted in a total of $n = 2000$ points. The approximated distribution was normalized such that area under the curve is 1. For the smoothing, we used the *KernelDensity* function in the *neighbors* module of the *Scikit-learn* Python package (**Pedregosa et al., 2011**). Details of the implementation can be found in the publicly available code repository. As with the Laplacian eigenspectrum, the spectral distance between every pair of animal species was estimated as 1 minus the cosine similarity of their smoothed (normalized) Laplacian eigenvalue distributions. Results of this supplementary analysis can be found in **Figure 2—figure supplement 9**.

## Topological distance

An alternative way to estimate inter-species distances in connectome organization is to compute the correlation between their network features. We estimated a set of local and global graph theory measures of the connectivity matrix. Local measures include node degree, clustering coefficient, node betweenness, and closeness. Global measures include characteristic path length, transitivity, and assortativity. We included both the binary and the weighted versions of these measures. We constructed a vector of local and global topological features for every animal species. Because there are as many local features as nodes in a network, we only used the average and the standard deviation of these measures. Similar to the spectral distance, the topological distance between every pair of animal species was estimated as 1 minus the cosine similarity of their topological feature vectors. All local and global features were estimated using the Python version of the Brain Connectivity Toolbox (https://github.com/aestrivex/bctpy; **Sporns et al., 2022**; **Rubinov and Sporns, 2010**). Definitions of these topological metrics can be found below.

### Local features

- *Degree (bin)*. Number of connections that a node participates in:

$$k_i = \sum_{j \neq i} a_{ij} \qquad \forall i \in N$$

  where $a_{ij} = 1$ if nodes $i$ and $j$ are connected, otherwise $a_{ij} = 0$. $N$ corresponds to the set of all nodes in the graph (**Rubinov and Sporns, 2010**).
- *Degree (wei)*. Sum of connection weights that a node participates in:

$$s_i = \sum_{j \neq i} w_{ij} \qquad \forall i \in N$$

  where $w_{ij}$ corresponds to the connection weight between nodes $i$ and $j$. $N$ corresponds to the set of all nodes in the graph (**Rubinov and Sporns, 2010**).
- *Clustering (bin)*. Proportion of transitive closures (closed triangles) around a node, that is, the fraction of neighbour nodes that are neighbours of each other:

$$c_i(A) \quad = \frac{\frac{1}{2} \sum_{j \neq i} \sum_{h \neq (i,j)} a_{ij} a_{ih} a_{jh}}{\frac{1}{2} k_i(k_i - 1)}$$

$$= \frac{(A^3)_{ii}}{k_i(k_i - 1)} \qquad \forall i \in N$$

where $A$ corresponds to the binary adjacency matrix of the graph, $(A^3)_{ii}$ is the ith element of the main diagonal of $A^3 = A \cdot A \cdot A$, and $ki$ is the degree of node $i$ (**Watts and Strogatz, 1998**; **Rubinov and Sporns, 2010**).

- *Clustering (wei)*. Mean 'intensity' of triangles around a node:

$$c_i(W) \quad = \frac{\frac{1}{2} \sum_{j \neq i} \sum_{h \neq (i,j)} w_{ij}^{\frac{1}{3}} w_{ih}^{\frac{1}{3}} w_{jh}^{\frac{1}{3}}}{\frac{1}{2} k_i (k_i - 1)}$$

$$= \frac{(W^{[\frac{1}{3}]})_{ii}^3}{k_i (k_i - 1)} \qquad \forall i \in N$$

where $W$ corresponds to the weighted connectivity matrix, $(W^{[\frac{1}{3}]})_{ii}^3$ is the ith element of the main diagonal of $(W^{[\frac{1}{3}]})^3 = W^{\frac{1}{3}} \cdot W^{\frac{1}{3}} \cdot W^{\frac{1}{3}}$, and $ki$ is the degree of node $i$ (**Onnela et al., 2005**; **Rubinov and Sporns, 2010**).

- *Shortest path length (bin)*. Minimum geodesic distance between pairs of nodes:

$$d_{ij} = \sum_{a_{uv} \in g_{i \leftrightarrow j}} a_{uv}$$

where $g_{i \leftrightarrow j}$ is the shortest geodesic path between nodes $i$ and $j$. Note that $d_{ij} = \infty$ for all disconnected pairs $i, j$ (**Rubinov and Sporns, 2010**).

- *Shortest path length (wei)*. Minimum weighted distance between pairs of nodes:

$$d_{ij}^{\text{w}} = \sum_{a_{uv} \in g_{i \leftrightarrow j}} f(w_{uv})$$

where $f$ is a map (e.g. the inverse) from weight to length and $g_{i \leftrightarrow j}$ is the shortest weighted path between nodes $i$ and $j$ (**Rubinov and Sporns, 2010**).

- *Betweenness (bin)*. Proportion of shortest (geodesic) paths in the graph that traverse a node:

$$b_i = \frac{1}{(n-1)(n-2)} \sum_{j \neq i} \sum_{h \neq (i,j)} \frac{\rho_{hj}(i)}{\rho_{hj}} \qquad \forall i \in N$$

where $\rho_{hj}$ is the number of shortest (geodesic) paths between nodes $h$ and $j$, $\rho_{hj}(i)$ is the number of shortest (geodesic) paths between nodes $h$ and $j$ that pass through $i$, and $n$ is the number of nodes in the graph (**Freeman, 1978**; **Brandes, 2001**; **Kintali, 2008**; **Rubinov and Sporns, 2010**).

- *Betweenness (wei)*. Same as *betweenness (bin)*, but shortest paths are estimated on the respective weighted graph (**Freeman, 1978**; **Brandes, 2001**; **Kintali, 2008**; **Rubinov and Sporns, 2010**).

- *Closeness (bin)*. Mean shortest (geodesic) path length from a node to all other nodes in the network:

$$e_i(A) = \frac{\sum_{j \neq i} \sum_{h \neq (i,j)} a_{ij} a_{ih} [d_{jh}(N_i)]^{-1}}{k_i (k_i - 1)} \qquad \forall i \in N$$

where $d_{jh}(N_i)$ is the shortest (geodesic) path length between nodes $j$ and $h$, which contains only neighbours of $i$, and is estimated on the corresponding binary adjacency matrix $A$ (**Latora and Marchiori, 2001**; **Rubinov and Sporns, 2010**).

- *Closeness (wei)*. Mean shortest (weighted) path from a node to all other nodes in the network:

$$e_i(W) = \frac{\sum_{j \neq i} \sum_{h \neq (i,j)} (w_{ij} w_{ih} [d_{jh}^{\text{w}}(N_i)]^{-1})^{\frac{1}{3}}}{k_i (k_i - 1)} \qquad \forall i \in N$$

where $d_{jh}^{\text{w}}(N_i)$ is the length of the shortest (weighted) path between nodes $j$ and $h$, which contains only neighbours of $i$, and is estimated on the corresponding weighted adjacency matrix $W$ (**Latora and Marchiori, 2001**; **Rubinov and Sporns, 2010**).

## Global features

- *Characteristic path length (bin)*. Average shortest (geodesic) path length between all pairs of nodes in the graph:

$$L = \frac{1}{n} \sum_{i \in N} \frac{\sum_{j \neq i} d_{ij}}{n-1}$$

where $d_{ij}$ is the shortest (geodesic) path length between nodes $i$ and $j$, and $n$ is the number of nodes in the graph (***Watts and Strogatz, 1998***; ***Rubinov and Sporns, 2010***).

- *Characteristic path length (wei)*. Average shortest (weighted) path length between all pairs of nodes in the graph:

$$L^{\mathrm{w}} = \frac{1}{n} \sum_{i \in N} \frac{\sum_{j \neq i} d_{ij}^{\mathrm{w}}}{n-1}$$

where $d_{ij}^{\mathrm{w}}$ is the shortest (weighted) path length between nodes $i$ and $j$, and $n$ is the number of nodes in the graph (***Watts and Strogatz, 1998***; ***Rubinov and Sporns, 2010***).

- *Transitivity (bin)*. Ratio between the observed number of closed triangles and the maximum possible number of closed triangles:

$$T(A) = \frac{\sum_{i \in N} \sum_{j,h \in N} a_{ij} a_{ih} a_{jh}}{\sum_{i \in N} k_i(k_i - 1)}$$

$$= \frac{\sum_{i \in N} (A^3)_{ii}}{\sum_{i \in N} k_i(k_i - 1)}$$

where $A$ is the binary adjacency matrix, and $k_i$ is the degree of node $i$ (***Newman, 2003***; ***Rubinov and Sporns, 2010***).

- *Transitivity (wei)*. Ratio between the 'intensity' of observed closed triangles and the maximum possible 'intensity' of closed triangles:

$$T(W) = \frac{\sum_{i \in N} \sum_{j,h \in N} w_{ij}^{\frac{1}{3}} w_{ih}^{\frac{1}{3}} w_{jh}^{\frac{1}{3}}}{k_i(k_i - 1)}$$

$$= \frac{\sum_{i \in N} (W^{[\frac{1}{3}]})_{ii}^3}{\sum_{i \in N} k_i(k_i - 1)}$$

where $W$ is the weighted connectivity matrix, and $k_i$ is the degree of node $i$ (***Newman, 2003***; ***Rubinov and Sporns, 2010***).

- *Assortativity (bin)*. Correlation coefficient between the degree of a node and the mean degree of its neighbours:

$$r = \frac{l^{-1} \sum_{(i,j) \in L} k_i k_j - [l^{-1} \sum_{(i,j) \in L} \frac{1}{2}(k_i + k_j)]^2}{l^{-1} \sum_{(i,j) \in L} \frac{1}{2}(k_i^2 + k_j^2) - [l^{-1} \sum_{(i,j) \in L} \frac{1}{2}(k_i + k_j)]^2}$$

where $l^{-1}$ is the inverse of the number of links in the graph, $k_i$ is the degree of node $i$, and $L$ is the set of links in the graph (***Newman, 2002***; ***Rubinov and Sporns, 2010***).

- *Assortativity (wei)*. Correlation coefficient between the weighted degree of a node and the mean weighted degree of its neighbour:

$$r^{\mathrm{w}} = \frac{l^{-1} \sum_{(i,j) \in L} w_{ij} s_i s_j - [l^{-1} \sum_{(i,j) \in L} \frac{1}{2} w_{ij}(s_i + s_j)]^2}{l^{-1} \sum_{(i,j) \in L} \frac{1}{2} w_{ij}(s_i^2 + s_j^2) - [l^{-1} \sum_{(i,j) \in L} \frac{1}{2} w_{ij}(s_i + s_j)]^2}$$

where $l^{-1}$ is the inverse of the number of links in the graph, $s_i$ is the weighted degree of node $i$, and $L$ is the set of links in the graph (***Leung and Chau, 2007***; ***Rubinov and Sporns, 2010***).

## Small-world organization

We use the index proposed in *Humphries and Gurney, 2008* to measure connectomes' small-worldness level. The index is given by

$$\gamma = \frac{C}{C_{rand}}$$

$$\lambda = \frac{L}{L_{rand}}$$

$$S = \frac{\gamma}{\lambda}$$

where $L$ and $C$ are the corresponding characteristic path length and clustering coefficient of each connectome, respectively, and $L_{rand}$ and $C_{rand}$ are the corresponding average quantities for a set of 1000 randomly rewired graphs that preserve the degree sequence and distribution of the nodes (*Maslov and Sneppen, 2002*). A network is said to possess a small-world architecture if $S \geq 1$, that is, if it situates above the identity line in a $\gamma$ vs. $\lambda$ plot.

## Multi-resolution community detection

We used the Louvain algorithm to determine the optimal community structure of connectomes (*Blondel et al., 2008*). Briefly, this algorithm extracts communities from large networks by optimizing a modularity score. Here, we use the Q-metric as the objective function (*Blondel et al., 2008*; *Fortunato and Barthélemy, 2007*):

$$Q(\gamma) = \frac{1}{2m} \sum_{ij} \left[ A_{ij} - \gamma \frac{k_i k_j}{2m} \right] \delta(c_i, c_j)$$

where $A$ corresponds to the adjacency matrix of the network, $k_i$ is the degree of node $i$, $m$ is the sum of all connections in the graph, $c_i$ is the community affiliation of node $i$, and $\delta$ is the Kronecker delta function (i.e. $\delta(x, y) = 1$ if $x = y$, 0 otherwise). The size of the partition is controlled by a resolution parameter $\gamma$ (higher $\gamma$ values result in a larger number of modules). Because the Louvain method is a greedy algorithm, we first found multiple (250) optimal partitions at $\gamma = 1$, and then we determined a single partition using a consensus clustering approach (*Bassett et al., 2013*).

## Classification of edges

Connectomes's connections were classified into different categories based on four criteria. The first criterion is based on the modular structure of the network and classifies connections depending on whether they link brain regions within the same module (i.e. intra-modular) or regions across different modules (i.e. inter-modular); the second criterion is whether connections link brain regions within the same hemisphere (i.e. intra-hemispheric connections) or across hemispheres (inter-hemispheric connections); the third criterion is based on the physical length of the connections (i.e. short-, medium-, and long-range connections); and the fourth criterion is based on the rich-club structure of the network (i.e. rich-club, feeder, and peripheral connections).

### Inter- vs. intra-modular connections

To classify connections as being either inter- or intra-modular, a consensus clustering algorithm was applied on each connectome to determine a partition of the network into different modules (see 'Community detection'). Once modules are identified, inter-modular connections correspond to those linking brain regions across different modules, whereas intra-modular connections correspond to those linking brain regions belonging to the same module.

### Connection length

Euclidean distance between regions' centres was used as a proxy for connection length. To subdivide connections into short-, medium-, and long-range, connection lengths were estimated as a percentage with respect to the maximum distance between regions. Short-range connections correspond to those that are less than or equal to 25% of the maximum distance; medium-range connections are above 25% but less than or equal to 75% of the maximum distance; and long-range connections are those above 75% of the maximum distance.

## Rich-club vs. feeder vs. peripheral connections

To classify edges as being either rich-club, feeder, or peripheral connections, it is necessary to identify first the rich-club of hubs in the network, that is, the densely interconnected core of nodes that have a disproportionately high number of connections (*van den Heuvel and Sporns, 2011*; *van den Heuvel et al., 2012*). To do so, we compute the rich-club coefficients $\Phi(k)$ across a range of degree $k$ of the unweighted (binary) connectomes. For binary networks, all nodes that show a degree $\leq k$ are removed from the network, and for the remaining set of nodes (i.e. a sub-graph), the rich-club coefficient is estimated as the ratio of connections present in the sub-graph, to the total number of possible connections that would be present if the resulting sub-graph was fully connected. Formally, the rich-club coefficient is given by *Zhou and Mondragon, 2004*; *Colizza et al., 2006*; *McAuley et al., 2007*

$$\phi(k) = \frac{2E_{>k}}{N_{>k}(N_{>k} - 1)}$$

In random networks, such as the Erdős–Rényi model, nodes with a higher degree have a higher probability of being interconnected by chance alone, thus showing an increasing function of $\Phi(k)$. For this reason, the rich-club coefficient is typically normalized relative to a set of $m$ comparable random networks of equal size and node degree sequence and distribution (*Maslov and Sneppen, 2002*; *Colizza et al., 2006*; *McAuley et al., 2007*). The normalized rich-club coefficient is then given by

$$\phi_{norm}(k) = \frac{\phi(k)}{\phi_{random}(k)}$$

where $\phi_{random}$ corresponds to the average rich-club coefficient over the $m$ random networks. In our particular case, $m = 1000$. An increasing normalized coefficient $\Phi_{norm} > 1$ over a range of $k$ reflects the existence of rich-club organization.

To assess the statistical significance of rich-club organization, we used permutation testing (*Bassett and Bullmore, 2009*; *van den Heuvel et al., 2010*). Briefly, the population of the $m$ random networks yields a null distribution of rich-club coefficients. For the range of $k$ expressing rich-club organization (i.e. $\Phi_{norm} > 1$), we tested whether $\phi(k)$ significantly exceeds $\Phi_{random}(k)$. Next, we identify the $k$th level at which the maximum significant $\Phi_{norm}$ occurs. Nodes with a degree $\geq k$ are said to belong to the $k$th-core of the network. Next, we identify the hubs, that is, nodes whose degree is above the average degree of the network plus 1 standard deviation. Therefore, rich-club nodes were identified as those nodes that are both hubs and belong to the $k$th-core of the network.

Once rich-club nodes are identified, rich-club connections are defined as edges between rich-club nodes; feeder connections are edges connecting rich-club to non-rich-club nodes, and peripheral connections are edges between non-rich-club nodes.

## Controlling for replicas

Because some of the species have multiple scans, this could bias the distribution of intra- and inter-order distances, which could be dominated by those species with a large number of replicas. To account for that, we randomly sample a single connectome per species, and we calculated inter-species distances. We repeated this procedure iteratively 10,000 times. The reported intra- and inter-order distance distributions correspond to the average distances across iterations.

## Controlling for density

Some of the graph features used for the estimation of the topological distance are highly dependent on the density of the network. To regress out the effects of network density on a graph feature, a univariate linear and an exponential model are fitted using density as the explanatory variable, and each feature as response variable. That is,

$$\text{Linear:} \quad y = ax + b$$

$$\text{Exponential:} \quad y = ae^{bx} + c$$

where $y$ represents network features and $x$ corresponds to density. Network features are then replaced by the residuals of the model. The decision to fit either a linear, an exponential, or no model at all was based on the variance explained by the model or $R^2$. The model with the largest

$R^2$ is selected. Only those features with $R^2 > 0.1$ were controlled to account for density (features in *Figure 3—figure supplement 5* with a regression line).

## Code availability

All codes used for data analysis and figure generation are publicly available on GitHub (https://github.com/netneurolab/suarez_connectometaxonomy; *Suarez, 2022* copy archived at swh:1:rev:0d8e-98f65a51a77784b31ec3ca59176d9119d927) and are built on top of the following open-source Python packages: rnns (https://github.com/estefanysuarez/rnns.git; *Suarez, 2021*), Netneurotools (https://github.com/netneurolab/netneurotools; *Markello et al., 2022*), Numpy (*Harris et al., 2020*; *van der Walt et al., 2011*; *Oliphant, 2006*), Scipy (*Virtanen et al., 2020*), Pandas (*McKinney, 2010*), Scikit-learn (*Pedregosa et al., 2011*), bctpy (https://github.com/aestrivex/bctpy; *Sporns et al., 2022*; *Rubinov and Sporns, 2010*), Matplotlib (*Hunter, 2007*), and Seaborn (*Waskom et al., 2016*).

## Acknowledgements

BM acknowledges support from the Natural Sciences and Engineering Research Council of Canada (NSERC), Canadian Institutes of Health Research (CIHR), Brain Canada Foundation Future Leaders Fund, the Canada Research Chairs (CRC) Program and the Healthy Brains for Healthy Lives (HBHL) initiative. GL acknowledges support from NSERC, the CRC Program, and the Canada CIFAR AI Cahir Program.

## Additional information

### Funding

| Funder | Grant reference number | Author |
|---|---|---|
| Natural Sciences and Engineering Research Council of Canada | | Bratislav Misic<br>Guillaume Lajoie |
| Canadian Institutes of Health Research | | Bratislav Misic |
| Fondation Brain Canada | Future Leaders Fund | Bratislav Misic |
| Canada Research Chairs | | Bratislav Misic<br>Guillaume Lajoie |
| Michael J. Fox Foundation for Parkinson's Research | | Bratislav Misic |
| Healthy Brains for Healthy Lives | | Bratislav Misic |
| Canadian Institute for Advanced Research | | Guillaume Lajoie |
| National Science Foundation - BSF | | Yaniv Assaf |
| National Institute of Mental Health | R01-MH122957 | Olaf Sporns |

The funders had no role in study design, data collection and interpretation, or the decision to submit the work for publication.

### Author contributions

Laura E Suarez, Conceptualization, Software, Formal analysis, Investigation, Methodology, Writing – original draft; Yossi Yovel, Yaniv Assaf, Data curation, Writing - review and editing; Martijn P van den Heuvel, Olaf Sporns, Writing - review and editing; Guillaume Lajoie, Conceptualization, Methodology, Writing - review and editing; Bratislav Misic, Conceptualization, Resources, Supervision, Funding acquisition, Validation, Visualization, Writing – original draft, Project administration

Author ORCIDs
Laura E Suarez http://orcid.org/0000-0003-0700-1500
Yaniv Assaf http://orcid.org/0000-0001-6941-1535
Bratislav Misic http://orcid.org/0000-0003-0307-2862

Decision letter and Author response
Decision letter https://doi.org/10.7554/eLife.78635.sa1
Author response https://doi.org/10.7554/eLife.78635.sa2

## Additional files

### Supplementary files
• MDAR checklist

### Data availability
The MaMI data set was originally collected and analyzed by Assaf and colleagues in *Assaf et al., 2020*. We have included the connectivity matrices used in this study in a public repository available at https://doi.org/10.5281/zenodo.7143143.

The following dataset was generated:

| Author(s) | Year | Dataset title | Dataset URL | Database and Identifier |
|---|---|---|---|---|
| Suarez LE, Yoval Y, van den Heuvel MP, Sporns O, Lajoie G, Misic B | 2022 | MaMI dataset | https://doi.org/10.5281/zenodo.7143143 | Zenodo, 10.5281/zenodo.7143143 |

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
