## [Editor Report]

This important article uses an impressively rich data set (obtained and curated by the authors) to compare the structural brain connectomes of many animals spanning six taxonomic orders. The approach is innovative and relies on graph theoretical measures to describe the connectivity, which means it can be done without the need to spatially/functionally match the brains. The authors find compelling evidence that there is more variability between than within order. They attribute this effect to changes in local connectivity features, whereas global patterns are preserved. The approach can potentially be a useful way to study phylogeny and brain evolution.

---

## [Decision Letter]

**Decision letter after peer review:**

Thank you for submitting your article "A connectomics-based taxonomy of mammals" for consideration by *eLife*, and sorry for the delay in getting the reviews back to you.

Your article has been reviewed by 3 peer reviewers, including Saad Jbabdi as Reviewing Editor and Reviewer #1, and the evaluation has been overseen by Chris Baker as the Senior Editor. The following individual involved in review of your submission has agreed to reveal their identity: Katja Heuer (Reviewer #3).

Essential revisions:

The paper was well received. There are several technical comments that need addressing, but the main revision in my opinion concerns the need for more insight as to what is driving the similarities and differences in the connectomes. See the detailed comments below.

Please find below the detailed reviews. I have tried to merge the 3 reviewer comments into a single set of comments, so these might not be in a sensible order.

Reviewer Comments

*Reviewers' (Recommendations for the authors):*

I think the assumption of the use of the same 200 regions for all species should be shown to be justified.

I would have liked to have seen more detail on how the differences between species is really implemented on the brains, if even in a few species. For instance, some of the authors have published extensively on human specializations in connectivity. It would be nice to show that some of the differences they identified in that work fall under the 'local regional connectivity profile' features that are reported here.

Species names sometimes correspond to groups of species. Would it be possible to provide the exact species names? The dataset in Zenodo includes, for example, "Macaque" which I assume may be Rhesus macaques, but for example, "colobus" is a group of monkeys including several species and it would be good to know which species have been included here. Given the vast nature of the dataset, either the exact English names, or the binomial species names, or a csv file which could allow potential users of the data to make the correspondence could be great to facilitate reuse of the data.

On page 6 the authors conclude that given the larger connectome similarity within one taxonomic group, the organisation of the brain connectome may be under selection pressure. However, also the Brownian motion model of evolution – where phenotypes are assumed to vary randomly along the phylogenetic tree – can explain phenotypes that are more similar within one taxonomic group.

When the authors introduce that the "common space" approach was used, maybe they can add a short note on what this means in addition to just the reference?

Methods: in the abstract, the authors say the data was "collected using a single protocol on a single scanner." whereas in the methods section they refer to 2 scanners and 2 field strengths. It seems quite standard to acquire small animal brains on a scanner with higher field strength and different head coils, there is no problem, I just thought maybe there is no need for the 1-scanner statement in the abstract? If the authors wanted to keep something along these lines, it could take up the idea they mention of a "unified MRI protocol was implemented for all species" I guess.

On page 7, the authors say "Consistent with the notion that neural circuit evolution involves local circuit modifications to adapt to specific challenges [10], such as extreme environmental pressures [36, 79, 89], or to support specific behaviours." I would just suggest a slight reformulation, as evolution doesn't do modifications *to adapt* to challenges or *to support* behaviour, but rather involves random modifications, that then may have provided an advantage in facing certain challenges or supporting certain behaviours.

On page 8, the authors suggest a "This scaling results in less space for white matter connectivity with increasing brain size." I wonder how this result compares with the reports of a positive allometry of white matter. For example, Zhang and Sejnowski

(https://doi.org/10.1073/pnas.090504197) report a scaling of 1.23 for white matter and build a scaling model for justifying it. There are several references within that paper that go in the same direction. The positive scaling of WM seems also intuitively true, as mice have very little WM compared with monkeys or humans, for example.

Other Comments:

– Visualisation

Some comments / suggestions on the various visualisations in the paper:

– I think a 2D scatter plot using multidimensional scaling would nicely show the between and within order distances (I tried it on your data, it looks nice). This could be complemented by a hierarchical clustering diagram?

– I am not sure why you need to min-max rescale all the distances to be between 0 and 1. In addition, min-max measures are sensitive to outliers. If one group happens to have an outlier, that could drive the entire group to be at the extreme. Is this likely to be happening here? It would also be useful to know at what stage this rescaling is done?

– I don't think the histograms in figures 2 and 3 should be shown using kde smoothing. It hides the data, and actually does a disservice to the data (e.g. in Figure 2B, the diagonal values are visibly lower than the extra diagonal values, but when kde-smoothed the effect appears to be lower than it is).

Another thing about the spectral approach: the lowest eigenvalue is famously important in telling us something about how connected the network is in general (i.e. how easy it is to break it down). Did you see any differences between and within species/orders in this measure?

---

## [Author Response]

Reviewers' (Recommendations for the authors):I think the assumption of the use of the same 200 regions for all species should be shown to be justified.

Please see public review for our response to the comment.

I would have liked to have seen more detail on how the differences between species is really implemented on the brains, if even in a few species. For instance, some of the authors have published extensively on human specializations in connectivity. It would be nice to show that some of the differences they identified in that work fall under the 'local regional connectivity profile' features that are reported here.

We concur with the Reviewer. A similar point was mentioned in the review summary – please see our response on pages 3-6. For convenience, we reproduce the response here.

The parcellations used in this study are data-driven and are not based on anatomical delineations of the cortex, which makes it hard to compare local connectivity profiles across species. This choice was based on two considerations: (1) for most species there exist no MRI parcellations, (2) in the few cases where such parcellations exist, there is incomplete knowledge about which regions (homologues) correspond to one another across species. In the original manuscript, the approach that we took to address this question was to examine how the difference between intra- and inter-order topological distances changes when different sets of topological features are included to estimate inter-species distances, , finding that local features differentiate species across taxonomic orders better than global features.

Nevertheless, we agree with the Reviewers that it is important to further investigate the nature of these local phylogenetic changes with greater anatomical specificity. Inspired by the Reviewers’ suggestion, we focused on a hallmark feature of network connectivity: the relative connectivity profiles of frontal cortex and occipital cortex. Numerous studies have suggested that the organization of the frontal cortex is more highly differentiated across species, whereas the organization of the visual and somatosensory cortices is more conserved [1-3]. Specifically, we selected the 10% most anterior and 10% most posterior nodes in each connectome as a proxy for frontal and occipital cortex. We then compared the binary degree, clustering, betweenness centrality and closeness of the anterior and posterior nodes (i.e. the average local features of those nodes) to assess how frontal and occipital connectivity profiles reconfigure across phylogeny. Figure 3 - figure supplement 2 shows the anterior-posterior differences for the six mammalian orders studied. The key finding is that mammalian orders are largely differentiated by the balance of local connectivity in anterior versus posterior cortex (separate one-way ANOVAs were performed to compare the effect of taxonomic order on anterior-posterior differences in local: *(i) degree*: F(5)=20.29, *p*=2.57x10^-16^*, (ii) clustering*: F(5)=15.88, *p*=3.88x10^-13^, *(iii) betweenness*: F(5)=17.97, *p*=1.14x10^-14^ and *(iv) closeness*: F(5)=13.54, *p*=2.36x10^-11^). The greatest difference in connectivity of the frontal cortex compared to the occipital cortex is observed in the Carnivora and Chiroptera orders, whereas the lowest difference is observed in Rodentia.

Even though there are differences in the balance between anterior and posterior local connectivity across taxonomic orders, we can observe a systematic trend for brain regions that are more anterior, or pertaining to the frontal cortex, to be more central, participating in a larger proportion of communication pathways (i.e. have greater degree and betweenness), and to engage in long-range connections to multiple other systems, presumably allowing them to sample information from multiple sensory domains. Conversely, regions that are more posterior, corresponding to visual cortex in most mammals, tend to have a larger proportion of short-range connections with a more densely clustered architecture, contrary to the more central architecture of anterior regions. These trends can be observed in the exemplars shown in Figure 3 —figure supplement 2, panel b.

Here, by showing just one of the many different ways in which local connectivity has reconfigured across phylogeny, we set the stage to keep developing new methods and analytical tools to explore ways in which local connectivity reconfigures during evolution.

These findings were added as supplementary analysis in Figure 3 —figure supplement 2, and they were introduced in the main text as the last sentence of the first paragraph of the *Architectural features differentiate species* subsection of the *Results section*. The new sentence reads:

“An illustration of this principle is depicted in Figure 3 —figure supplement 2 showing that the relative local connectivity of frontal and occipital cortices changes across taxonomic orders [11, 58, 59].”

[11] Barrett, R. L., Dawson, M., Dyrby, T. B., Krug, K., Ptito, M., D'Arceuil, H., … and Catani, M. (2020). Differences in frontal network anatomy across primate species. *Journal of Neuroscience*, *40*(10), 2094-2107.

[58] Krubitzer, L., and Kaas, J. (2005). The evolution of the neocortex in mammals: how is phenotypic diversity generated?. *Current opinion in neurobiology*, *15*(4), 444-453.

[59] Krubitzer, L., and Kahn, D. M. (2003). Nature versus nurture revisited: an old idea with a new twist. *Progress in neurobiology*, *70*(1), 33-52.

In addition, this suggestion from the Reviewers encouraged us to add further discussion and perspective on how future comparative studies can make progress in resolving homologies. The following new paragraph was added after the third paragraph of the Discussion section:

“These results highlight the importance of developing species-specific, anatomical based parcellations, as well as new ways to align connectomes from different species. Understanding how homologous regions correspond to one another will allow further investigation of regional inter-species differences in connectome topology, which is a fundamental step for advancing comparative connectomics. A variety of emerging methods are contributing to further resolve correspondence between brain regions across species, facilitating fine-grained comparisons at the level of individual regions. These methods implement regional comparisons based on different data modalities including measures of cytoarchitecture [21, 46], receptor distribution [64], functional and structural connectivity fingerprints [74, 75, 91], patterns of gene expression [18, 120], and macro scale gradients of functional activation [26].”

[18] Beauchamp, A., Yee, Y., Darwin, B., Raznahan, A., Mars, R. B., and Lerch, J. P. (2022). Whole-brain comparison of rodent and human brains using spatial transcriptomics. *bioRxiv*.

[21] Bianchi, S., Stimpson, C. D., Bauernfeind, A. L., Schapiro, S. J., Baze, W. B., McArthur, M. J., … and Sherwood, C. C. (2013). Dendritic morphology of pyramidal neurons in the chimpanzee neocortex: regional specializations and comparison to humans. *Cerebral cortex*, *23*(10), 2429-2436.

[26] Buckner, R. L., and Margulies, D. S. (2019). Macroscale cortical organization and a default-like apex transmodal network in the marmoset monkey. *Nature communications*, *10*(1), 1-12.

[46] Garey, L. J. (Ed.). (1999). *Brodmann's' localisation in the cerebral cortex'*. World Scientific.

[64] Levant, B. (1998). Differential distribution of D3 dopamine receptors in the brains of several mammalian species. *Brain research*, *800*(2), 269-274.

[74] Mars, R. B., Sotiropoulos, S. N., Passingham, R. E., Sallet, J., Verhagen, L., Khrapitchev, A. A., … and Jbabdi, S. (2018). Whole brain comparative anatomy using connectivity blueprints. *eLife*, *7*, e35237.

[75] Mars, R. B., Verhagen, L., Gladwin, T. E., Neubert, F. X., Sallet, J., and Rushworth, M. F. (2016). Comparing brains by matching connectivity profiles. *Neuroscience and Biobehavioral Reviews*, *60*, 90-97.

[91] Passingham, R. E., Stephan, K. E., and Kötter, R. (2002). The anatomical basis of functional localization in the cortex. *Nature Reviews Neuroscience*, *3*(8), 606-616.

[120] Warrington, S., Thompson, E., Bastiani, M., Dubois, J., Baxter, L., Slater, R., … and Sotiropoulos, S. N. (2022). Concurrent mapping of brain ontogeny and phylogeny within a common connectivity space. *bioRxiv*.

References used in the response:

[1] Barrett, R. L., Dawson, M., Dyrby, T. B., Krug, K., Ptito, M., D'Arceuil, H., … and Catani, M. (2020). Differences in frontal network anatomy across primate species. *Journal of Neuroscience*, *40*(10), 2094-2107.

[2] Krubitzer, L., and Kahn, D. M. (2003). Nature versus nurture revisited: an old idea with a new twist. *Progress in neurobiology*, *70*(1), 33-52.

[3] Krubitzer, L., and Kaas, J. (2005). The evolution of the neocortex in mammals: how is phenotypic diversity generated?. *Current opinion in neurobiology*, *15*(4), 444-453.

Species names sometimes correspond to groups of species. Would it be possible to provide the exact species names? The dataset in Zenodo includes, for example, "Macaque" which I assume may be Rhesus macaques, but for example, "colobus" is a group of monkeys including several species and it would be good to know which species have been included here. Given the vast nature of the dataset, either the exact English names, or the binomial species names, or a csv file which could allow potential users of the data to make the correspondence could be great to facilitate reuse of the data.

We agree with the Reviewer on this point and we have updated Figure 1 —figure supplement 1 in the main manuscript, as well as the file “info.csv” (in the public data repository of the manuscript – url:10.5281/zenodo.6376543) to include the specific species’ names.

On page 6 the authors conclude that given the larger connectome similarity within one taxonomic group, the organisation of the brain connectome may be under selection pressure. However, also the Brownian motion model of evolution – where phenotypes are assumed to vary randomly along the phylogenetic tree – can explain phenotypes that are more similar within one taxonomic group.

We added a sentence to suggest that there exist alternative explanations for the forces driving phenotype evolution based on the Brownian motion model of evolution. The new sentence reads:

“Specifically, species that are part of the same taxonomic order tend to display similar connectome architecture, suggesting that brain network organization is under selection pressure (see also [29, 60, 126] for an alternative mechanism of trait evolution characterized by pure drift models based on Brownian motion), analogous to size, weight or color.”

[29] Butler, M. A., and King, A. A. (2004). Phylogenetic comparative analysis: a modeling approach for adaptive evolution. *The American Naturalist*, *164*(6), 683-695.

[60] Lande, R. (1976). Natural selection and random genetic drift in phenotypic evolution. *Evolution*, 314-334.

[126] Wright, S. (1931). Evolution in Mendelian populations. *Genetics*, *16*(2), 97.

When the authors introduce that the "common space" approach was used, maybe they can add a short note on what this means in addition to just the reference?

We followed the Reviewer’s advice and we have revised the paragraph to clarify what we mean by the term “common space”. The new paragraph reads:

“To identify brain connectivity differences across species, we need to be able to analyze data in a shared frame of reference. The normalized Laplacian eigenspectrum and the graph features of the connectivity matrix allow us to translate connectomes into a common feature space in which they are comparable, despite the fact that they come from different species, and that the nodes do not correspond to one another [72].”

[72] Mars, R. B., Jbabdi, S., and Rushworth, M. F. (2021). A common space approach to comparative neuroscience. *Annual Review of Neuroscience*, *44*.

Methods: in the abstract, the authors say the data was "collected using a single protocol on a single scanner." whereas in the methods section they refer to 2 scanners and 2 field strengths. It seems quite standard to acquire small animal brains on a scanner with higher field strength and different head coils, there is no problem, I just thought maybe there is no need for the 1-scanner statement in the abstract? If the authors wanted to keep something along these lines, it could take up the idea they mention of a "unified MRI protocol was implemented for all species" I guess.

We agree with the Reviewer that there is no need for the 1-scanner statement in the Abstract. We have rephrased this sentence in the Abstract to read:

“We analyze the mammalian MRI (MaMI) data set, a database that encompasses high-resolution ex vivo structural and diffusion magnetic resonance imaging (MRI) scans of 124 species across 12 taxonomic orders and 5 superorders, collected using a unified MRI protocol.”

On page 7, the authors say "Consistent with the notion that neural circuit evolution involves local circuit modifications to adapt to specific challenges [10], such as extreme environmental pressures [36, 79, 89], or to support specific behaviours." I would just suggest a slight reformulation, as evolution doesn't do modifications to adapt to challenges or to support behaviour, but rather involves random modifications, that then may have provided an advantage in facing certain challenges or supporting certain behaviours.

We agree with the Reviewer that the way this sentence is written misrepresents the way evolution is thought to operate. We have revised the manuscript and the new sentence reads:

“Along the same lines, our results are also consistent with the notion that neural circuit evolution involves random local circuit modifications that may have provided species with behavioral adaptations, allowing them to face specific challenges [10], such as extreme environmental pressures [41, 90, 101], or to support specific behaviours, such as courtship [9, 39, 40, 56, 71, 88, 99, 129], social bonding [54, 55, 66, 124], or foraging [89, 117].”

[9] Barkan, C. L., Kelley, D. B., and Zornik, E. (2018). Premotor neuron divergence reflects vocal evolution. *Journal of Neuroscience*, *38*(23), 5325-5337.

[10] Barker, A. J. (2021). Brains and speciation: Control of behavior. *Current Opinion in Neurobiology*, *71*, 158-163.

[39] Ding, Y., Berrocal, A., Morita, T., Longden, K. D., and Stern, D. L. (2016). Natural courtship song variation caused by an intronic retroelement in an ion channel gene. *Nature*, *536*(7616), 329-332.

[40] Ding, Y., Lillvis, J. L., Cande, J., Berman, G. J., Arthur, B. J., Long, X., … and Stern, D. L. (2019). Neural evolution of context-dependent fly song. *Current biology*, *29*(7), 1089-1099.

[41] Eigenbrod, O., Debus, K. Y., Reznick, J., Bennett, N. C., Sánchez-Carranza, O., Omerbašić, D., … and Lewin, G. R. (2019). Rapid molecular evolution of pain insensitivity in multiple African rodents. *Science*, *364*(6443), 852-859.

[54] Insel, T. R., and Shapiro, L. E. (1992). Oxytocin receptor distribution reflects social organization in monogamous and polygamous voles. *Proceedings of the National Academy of Sciences*, *89*(13), 5981-5985.

[55] Jaggard, J. B., Lloyd, E., Yuiska, A., Patch, A., Fily, Y., Kowalko, J. E., … and Keene, A. C. (2020). Cavefish brain atlases reveal functional and anatomical convergence across independently evolved populations. *Science advances*, *6*(38), eaba3126.

[56] Khallaf, M. A., Auer, T. O., Grabe, V., Depetris-Chauvin, A., Ammagarahalli, B., Zhang, D. D., … and Knaden, M. (2020). Mate discrimination among subspecies through a conserved olfactory pathway. *Science advances*, *6*(25), eaba5279.

[66] Loomis, C., Peuß, R., Jaggard, J. B., Wang, Y., McKinney, S. A., Raftopoulos, S. C., … and Duboue, E. R. (2019). An adult brain atlas reveals broad neuroanatomical changes in independently evolved populations of Mexican cavefish. *Frontiers in neuroanatomy*, *13*, 88.

[71] Markow, T. A., and O'Grady, P. M. (2005). Evolutionary genetics of reproductive behavior in *Drosophila*: connecting the dots. *Annual review of genetics*, *39*(1), 263-291.

[88] O’Grady, P. M., and DeSalle, R. (2018). Phylogeny of the genus *Drosophila*. *Genetics*, *209*(1), 1-25.

[89] Pantoja, C., Larsch, J., Laurell, E., Marquart, G., Kunst, M., and Baier, H. (2020). Rapid effects of selection on brain-wide activity and behavior. *Current Biology*, *30*(18), 3647-3656.

[90] Park, T. J., Lu, Y., Jüttner, R., Smith, E. S. J., Hu, J., Brand, A., … and Lewin, G. R. (2008). Selective inflammatory pain insensitivity in the African naked mole-rat (Heterocephalus glaber). *PLoS biology*, *6*(1), e13.

[99] Seeholzer, L. F., Seppo, M., Stern, D. L., and Ruta, V. (2018). Evolution of a central neural circuit underlies *Drosophila* mate preferences. *Nature*, *559*(7715), 564-569.

[101] Smith, E. S. J., Omerbašić, D., Lechner, S. G., Anirudhan, G., Lapatsina, L., and Lewin, G. R. (2011). The molecular basis of acid insensitivity in the African naked mole-rat. *Science*, *334*(6062), 1557-1560.

[117] Vanwalleghem, G. C., Ahrens, M. B., and Scott, E. K. (2018). Integrative whole-brain neuroscience in larval zebrafish. *Current opinion in neurobiology*, *50*, 136-145.

[124] Winslow, J. T., Hastings, N., Carter, C. S., Harbaugh, C. R., and Insel, T. R. (1993). A role for central vasopressin in pair bonding in monogamous prairie voles. *Nature*, *365*(6446), 545-548.

[129] York, R. A., Byrne, A., Abdilleh, K., Patil, C., Streelman, T., Finger, T. E., and Fernald, R. D. (2019). Behavioral evolution contributes to hindbrain diversification among Lake Malawi cichlid fish. *Scientific reports*, *9*(1), 1-9.

On page 8, the authors suggest a "This scaling results in less space for white matter connectivity with increasing brain size." I wonder how this result compares with the reports of a positive allometry of white matter. For example, Zhang and Sejnowski(https://doi.org/10.1073/pnas.090504197) report a scaling of 1.23 for white matter, and build a scaling model for justifying it. There are several references within that paper that go in the same direction. The positive scaling of WM seems also intuitively true, as mice have very little WM compared with monkeys or humans, for example.

The sentence cited here by the Reviewer makes reference to the negative allometric scaling results reported in [1] between cortical surface area and the total volume devoted to cerebral white matter. In contrast, the observed positive allometric scaling exponents reported in [2] and other studies [3-5] are between white matter volume and gray matter volume. These results are complementary rather than contradictory.

To maintain connectivity,arger brains require longer fibers to communicate between distant cortical areas. This is supported by the positive allometric scaling reported in [2] between white matter volume and gray matter volume, which indicates that the volume of the white matter that comprises long axons increases disproportionately faster compared to the volume of the gray matter corresponding to cell bodies, dendrites and axons for local information processing. On the other hand, the negative allometric scaling reported in [1] between cortical surface area and white matter volume suggests that, while the proportion of total volume devoted to cerebral white matter is higher in larger brains, it does not keep pace with the rapid cortical expansion that occurs with larger brain size. Therefore, as it is mentioned in the sentence cited by the Reviewer, there is less and less space available for white matter connectivity with increasing brain size, which translates into larger brains with a relatively higher proportion of short-range connections than long-range connections when compared with smaller brains. Synthesizing these results, we conclude that, despite the higher proportion -in comparison to gray matter-, white matter volume is relatively smaller in larger brains compared to smaller brains.

To avoid confusion, we have revised the manuscript and added the following sentence to clarify that these results are not in contradiction with previous studies. We:

“These results, however, do not contradict studies showing a positive allometric scaling between white matter and gray matter volume [44, 97, 109, 131]; while the proportion of total volume devoted to cerebral white matter is higher in larger brains, it does not keep pace with the rapid cortical expansion that occurs with larger brain size.”

[44] Frahm, H. D., Stephan, H., and Stephan, M. (1982). Comparison of brain structure volumes in Insectivora and Primates. I. Neocortex. *Journal fur Hirnforschung*, *23*(4), 375-389.

[97] Schlenska, G. (1974). Volumen-und oberflachenmessungen an gehirnen verschiedener saugetiere im vergleich zu einem errechneten modell.

[109] Theunissen, B. (1989). The Debate. In *Eugène Dubois and the Ape-Man from Java* (pp. 79-127). Springer, Dordrecht.

[131] Zhang, K., and Sejnowski, T. J. (2000). A universal scaling law between gray matter and white matter of cerebral cortex. *Proceedings of the National Academy of Sciences*, *97*(10), 5621-5626.

References used in the response:

[1] Ardesch, D. J., Scholtens, L. H., De Lange, S. C., Roumazeilles, L., Khrapitchev, A. A., Preuss, T. M., … and Van Den Heuvel, M. P. (2022). Scaling principles of white matter connectivity in the human and nonhuman primate brain. *Cerebral Cortex*, *32*(13), 2831-2842.

[2] Zhang, K., and Sejnowski, T. J. (2000). A universal scaling law between gray matter and white matter of cerebral cortex. *Proceedings of the National Academy of Sciences*, *97*(10), 5621-5626.

[3] Theunissen, B. (1989). The Debate. In *Eugène Dubois and the Ape-Man from Java* (pp. 79-127). Springer, Dordrecht.

[4] Schlenska, G. (1974). Volumen-und oberflachenmessungen an gehirnen verschiedener saugetiere im vergleich zu einem errechneten modell.

[5] Frahm, H. D., Stephan, H., and Stephan, M. (1982). Comparison of brain structure volumes in Insectivora and Primates. I. Neocortex. *Journal fur Hirnforschung*, *23*(4), 375-389.

Other Comments:– VisualisationSome comments / suggestions on the various visualisations in the paper:– I think a 2D scatter plot using multidimensional scaling would nicely show the between and within order distances (I tried it on your data, it looks nice). This could be complemented by a hierarchical clustering diagram?

We appreciate the Reviewer’s suggestion, and we have added as supplementary figures the 2D scatter plots using multidimensional scaling (Figure 3 —figure supplement 7), as well as the hierarchical clustering diagrams (Figure 3 —figure supplement 8). For the 2D projection of the data, we used all data samples including replicas, whereas for the hierarchical clustering algorithm we used the average inter-species distance matrix obtained after randomly resampling one sample per species.

We added a new paragraph at the end of the *Architectural features differentiate species* subsection of the *Results section* to introduce these new supplementary analyses. The new paragraph reads:

“Altogether, our results show that the subset of features that best differentiate species across taxonomic orders are the binary local topological features. We perform a set of complementary analyses to assess which subset of features produces the best partition of animal species relative to traditional taxonomies. To do so, we (a) project the data on a 2D plane using multidimensional scaling (Figure 3 —figure supplement 7), and (b) apply hierarchical clustering to inter-species distance matrices (Figure 3 —figure supplement 8). Visual inspection of these results suggests that, consistent with our previous results (Figure 3), local features compared to global features (ignoring panel a, center vs right column, respectively, in Figure 3 —figure supplement 7 and Figure 3 —figure supplement 8), as well as binary features compared to weighted features (ignoring panel a, center vs bottom row, respectively, in Figure 3 —figure supplement 7 and Figure 3 —figure supplement 8), yield species partitions that more closely reflect established phylogenetic relationships, further supporting the idea that connectome organization recapitulates traditional taxonomic relationships that are based on morphology and genetics.”

– I am not sure why you need to min-max rescale all the distances to be between 0 and 1. In addition, min-max measures are sensitive to outliers. If one group happens to have an outlier, that could drive the entire group to be at the extreme. Is this likely to be happening here? It would also be useful to know at what stage this rescaling is done?

Min-max scaling was applied at the last stage for visualization purposes, since scaling distance values between 0 and 1 facilitates the interpretation of the results. Because min-max scaling is linear, relative *distances* between (inter-species distance) values are preserved after scaling and hence neither the results are altered in any way, nor are they sensitive to outliers. Author response image 1 shows the same results as in Figure 2 without applying min-max scaling. Because interpreting and comparing values is easier when the minimum and maximum reference values are 0 and 1, respectively, we decided to keep the scaling to show the results in the manuscript.

**Author response image 1. sa2fig1:** Inter-species spectral distance without min-max scaling.

– I don't think the histograms in figures 2 and 3 should be shown using kde smoothing. It hides the data, and actually does a disservice to the data (e.g. in Figure 2B, the diagonal values are visibly lower than the extra diagonal values, but when kde-smoothed the effect appears to be lower than it is).

We followed the Reviewer’s suggestion to display the distributions in Figure 2 and Figure 3 as histograms, instead of using kde smoothing. Figure 2 and Figure 3 in the original manuscript were replaced by revised Figure 2 and Figure 3.

Another thing about the spectral approach: the lowest eigenvalue is famously important in telling us something about how connected the network is in general (i.e. how easy it is to break it down). Did you see any differences between and within species/orders in this measure?

It has been established theoretically that the smallest eigenvalue of the normalized Laplacian is necessarily zero regardless of the topology of the graph, i.e. λ1=0 [1]. We verified that for all samples that was actually the case. The second eigenvalue, however, indicates the connectedness of the graph. Specifically, if λ2>0, then the graph is connected [1]. Because connectomes represent brain networks, we would expect that this was the case for all animal species, otherwise this would suggest that there are disconnected or isolated brain regions. We examined the second eigenvalue of the Laplacian eigenspectra, and we found that for all species it is the case that λ2>0, except for four samples belonging to the taxonomic order *Chiroptera,* namely the Artibeus Jamacien, Myotis Emargenitus, Pkuhlii (3) and Tadarida Teniotis (1), which had 2/200, 5/200, 2/200 and 2/200 disconnected nodes, respectively. Despite the fact that these four connectomes have disconnected components, they were still included in the analyses but care was taken when estimating global topological features that might have been affected by this, such as *shortest path length (wei)* and *shortest path length (bin)* (by invoking function arguments that ignore infinite-length paths); local features are not significantly impacted since the average across the 200 nodes was used. Figure R5 shows the distribution of the second eigenvalue across taxonomic orders.

**Author response image 2. sa2fig2:** Distribution of the second eigenvalue of the (normalized) Laplacian eigenspectrum across taxonomic orders.

References used in the response:

[1] Chung, F. R., and Graham, F. C. (1997). *Spectral graph theory* (Vol. 92). American Mathematical Soc..